# Detectability of Arctic methane sources at six sites performing continuous atmospheric measurements

**Thibaud Thonat[1], Marielle Saunois[1], Philippe Bousquet[1], Isabelle Pison[1], Zeli Tan[2], Qianlai Zhuang[3], Patrick M. Crill[4], Brett F. Thornton[4], David Bastviken[5], Ed J. Dlugokencky[6], Nikita Zimov[7], Tuomas Laurila[8], Juha Hatakka[9], Ove Hermansen[9], and Doug E. J. Worthy[10]**

[1] Laboratoire des Sciences du Climat et de l'Environnement, LSCE/IPSL, CEA-CNRS-UVSQ, Université Paris-Saclay, F-91191 Gif-sur-Yvette, France
[2] Pacific Northwest National Laboratory, Richland, Washington, USA
[3] Department of Earth, Atmospheric, and Planetary Sciences, Purdue University, West Lafayette, Indiana, USA
[4] Department of Geological Sciences and Bolin Centre for Climate Research, Svante Arrhenius väg 8, 106 91, Stockholm, Sweden
[5] Department of Thematic Studies – Environmental Change, Linköping University, 581 83 Linköping, Sweden
[6] NOAA Earth System Research Laboratory, Global Monitoring Division, Boulder, Colorado, USA
[7] Northeast Science Station, Cherskiy, Russia
[8] Climate and Global Change Research, Finnish Meteorological Institute, Helsinki, Finland
[9] NILU – Norwegian Institute for Air Research, Kjeller, Norway
[10] Environment Canada, Toronto, Ontario, Canada

**Abstract**. Understanding the recent evolution of methane emissions in the Arctic is necessary to interpret the global methane cycle. Emissions are affected by significant uncertainties and are sensitive to climate change, leading to potential feedbacks. A polar version of the CHIMERE chemistry-transport model is used to simulate the evolution of tropospheric methane in the Arctic during 2012, including all known regional anthropogenic and natural sources, in particular freshwater emissions which are often overlooked in methane modelling. CHIMERE simulations are compared to atmospheric continuous observations at six measurement sites in the Arctic region. In winter, the Arctic is dominated by anthropogenic emissions; emissions from continental seepages and oceans, including from the East Siberian Arctic Shelf, can contribute significantly in more limited areas. In summer, emissions from wetland and freshwater sources dominate across the whole region. The model is able to reproduce the seasonality and synoptic variations of methane measured at the different sites. We find that all methane sources significantly affect the measurements at all stations at least at the synoptic scale, except for biomass burning. In particular, freshwaters play a decisive part in summer, representing on average between 11 and 26% of the simulated Arctic methane signal at the sites. This indicates the relevance of continuous observations to gain a mechanistic understanding of Arctic methane sources. Sensitivity tests reveal that the choice of the land surface model used to prescribe wetland emissions can be critical in correctly representing methane mixing ratios. The closest agreement with the observations is reached when using the two wetland models whose emissions peak in August-September, while all others reach their maximum in June-July. Such phasing provides an interesting constraint on wetland models which still have large uncertainties at present. Also testing different freshwater emission inventories leads to large differences in modelled methane. Attempts to include methane sinks (OH oxidation and soil uptake) reduced the model bias relative to observed atmospheric methane. The study illustrates how multiple sources, having different

spatiotemporal dynamics and magnitudes, jointly influence the overall Arctic methane budget, and highlights ways towards further improved assessments.

## 1 Introduction

The climate impact of atmospheric methane ($CH_4$) makes it the second most important anthropogenic greenhouse gas, being responsible for about one fifth of the total increase in radiative forcing since pre-industrial times. Since then, its concentration has increased by about 150% (IPCC, 2013). Between 1999 and 2006, the atmospheric methane burden remained nearly constant (Dlugokencky et al., 2009). The attribution of the cause of the 60 renewed rise after 2006 is still widely debated (e.g., Nisbet et al., 2014). A number of different processes have been examined including changes in anthropogenic sources (Schaefer et al., 2016; Hausmann et al., 2016; Schwietzke et al., 2016), in natural wetlands (Bousquet et al., 2011; Nisbet et al., 2016, McNorton et al., 2016), or in methane lifetime (Dalsøren et al., 2016; Rigby et al., 2017; Turner et al., 2017).

Recent changes in methane concentrations are not uniform and vary with latitude. The rise in methane in 2007 was, for example, particularly important in the Arctic region due to anomalously high temperatures leading to high wetland emissions (Dlugokencky et al., 2011; Bousquet et al., 2011). The Arctic ($> 60°N$) is of particular interest given the size of its carbon 70 reservoirs and the amplitude of recent and projected climate changes. It sequesters about 50% of the global organic soil carbon (Tarnocai et al., 2009). Decomposition of its most superficial fraction can lead to important feedbacks to climate warming. The Arctic is already affected by an amplification of climate warming; warming there is about twice that of the rest of the world (Christensen et al., 2013). Between 1950 and 2012, combined land and sea-surface 75 mean temperature had increased by about 1.6 °C in the region (AMAP, 2015), and climate projections predict temperature changes of a few degrees over the next decades (Collins et al., 2013). The Arctic represents now about 4% of the global methane budget (23 vs. 568 TgCH$_4$ yr$^{-1}$ for 2012, according to Saunois et al. (2016)). This budget is lower than bottom-up estimates (range 37-89 TgCH$_4$ yr$^{-1}$, according to the review by Thornton et al. 80 (2016b)), which are affected by large uncertainties. Although there is no sign of dramatic permafrost carbon emissions yet (Walter Anthony et al., 2016), thawing permafrost could double 21$^{st}$ century's Arctic methane budget and impact climate for centuries (Schuur et al. 2015).

85 This context points to the need for closely monitoring Arctic sources. The largest individual natural source from high latitudes is wetlands. An ensemble of process-based land surface models indicate that, between 2000 and 2012, wetland emissions have increased in boreal regions by 1.3 TgCH$_4$, possibly due to increases in wetland area and in air temperature (Poulter et al., submitted). However, different models show large discrepancies (model spread 90 of 80 TgCH$_4$ yr$^{-1}$ globally) even when using the same wetland emitting areas. Furthermore, the seasonality of Arctic natural continental emissions has been questioned, in particular by Zona et al. (2016), who suggested significant winter emissions from drier areas when soil temperatures are poised near 0°C. Significant methane enhancements have been observed in late fall/early winter in the Alaska North Slope (Sweeney et al., 2016) and in Greenland 95 (Mastepanov et al., 2008), where they were linked to Arctic tundra emissions, and also during spring thaw of shallow lakes (Jammet et al., 2015).

Freshwater emissions are another important and uncertain terrestrial source of methane. About 40% of the world's lakes are located north of 45°N (Walter et al., 2007) and their

emissions are expected to increase under a warming climate (Wik et al., 2016). Estimates for the high latitudes, extrapolated from measurements from different samples of lakes can vary from 13.4 $TgCH_4$ $yr^{-1}$ (above 54°N, Bastviken et al. (2011)) to 24.2 $TgCH_4$ $yr^{-1}$ (above 45°N, Walter et al. (2007)). Based upon a synthesis of 733 measurements made in Scandinavia, Siberia, Canada and Alaska, Wik et al. (2016) have assessed emissions north of 50°N at 16.5 $TgCH_4$ $yr^{-1}$. They have also highlighted the emissions' dependence on the water body type. Using a process-based lake biogeochemical model, Tan and Zhuang (2015a) have come to an estimate of 11.9 $TgCH_4$ $yr^{-1}$ north of 60°N, in the range of previous studies. This important source is generally poorly or not represented in large-scale atmospheric studies (Kirschke et al., 2013).

Additional continental sources include anthropogenic emissions, mostly from Russian fossil fuel industries, and, to a lesser extent, biomass burning, mostly originating from boreal forest fires. The Arctic is also under the influence of transported emissions from mid-latitudes methane sources, mostly of human origin (e. g., Paris et al., 2010; Law et al., 2014).

Marine emissions from the Arctic Ocean are smaller than terrestrial emissions, but they too are climate sensitive and affected by large uncertainties. Sources within the ocean include emissions from geological seeps, from sediment biology, from underlying thawing permafrost or hydrates, and from production in surface waters (Kort et al., 2012). The East Siberian Arctic Shelf (ESAS, in the Laptev and East Siberian Seas), which comprises more than a quarter of the Arctic shelf (Jakobsson et al., 2002) and most of subsea permafrost (Shakhova et al., 2010), is a large reservoir of carbon and most likely the biggest emission area (McGuire et al., 2009). Investigations led by Shakhova et al. (2010, 2014) estimated total ESAS emissions from diffusion, ebullition and storm-induced degassing, at 8−17 $TgCH_4$ $yr^{-1}$. A subsequent measurement campaign led by Thornton et al. (2016a), though not made during a stormy period, failed to observe the high rates of continuous emissions reported by Shakhova et al. (2014), and instead estimated an average flux of 2.9 $TgCH_4$ $yr^{-1}$. Berchet et al. (2016) also found that such values were not supported by atmospheric observations, and suggested instead the range of 0.0–4.5 $TgCH_4$ $yr^{-1}$.

The main sink of methane is its reaction with the hydroxyl radical (OH) in the troposphere, which explains about 90% of its loss. Other tropospheric losses include reaction with atomic chlorine (Cl) in the marine boundary layer (Allan et al., 2007) and oxidation in soils (Zhuang et al., 2013). These sinks vary seasonally, especially in the Arctic atmosphere, and their intensity is at their maximum in summer, when Arctic emissions are the highest. A good representation of the methane budget thus requires a proper knowledge of these sinks.

As mentioned before, a better understanding of methane sources and sinks and of their variations is critical in the context of climate change. Methane emissions can be estimated either by bottom-up studies, relying on extrapolation of flux measurements, on inventories and process-based models, or by top-down inversions which optimally combine atmospheric observations, transport modelling and a prior knowledge of emissions and sinks. The main input for top-down inversions is measurements of atmospheric methane mixing ratios, either at the surface or from space. Such observations are critical and should be made over long time periods to assess trends and variability. Surface methane monitoring started in the Arctic in the mid-1980s. Although more than 15 sites currently exist, six of them being in continuous operation (in addition to tower sites such as the JR-STATION tower network over Siberia (Sasakawa et al., 2010)), the observational network remains limited considering the Arctic area and the variety of existing sources (AMAP, 2015).

Retrievals of methane concentrations have been made from space since the mid-2000s, from global and continuous observations. However, in high latitudes, passive spaceborne sounders are limited by the availability of clear-sky spots and by sunlight (for NIR/SWIR instruments), and have been affected by persistent biases (e.g., Alexe et al., 2015; Locatelli et al., 2015). This is why only surface measurements, which provide precise and accurate data, are used in this study.

One interesting feature of Arctic methane emissions is that they are generally more distinct spatially and temporally (no or low wetland emissions in winter; anthropogenic emissions all year round) as compared to tropical emissions (e.g., in Northern India). Also, fast horizontal winds more efficiently relate emissions to atmospheric measurements (e.g., Berchet et al., 2016).

Methane modelling studies that rely on Arctic measurements have been used, for example, to assess the sensitivity of Arctic methane concentrations to uncertainties in its sources, in particular concerning the seasonality of wetland emissions and the intensity of ESAS emissions (Warwick et al., 2016; Berchet et al., 2016). Top-down inversions have also led to methane surface flux estimates and discussions of their variations. For instance, Thompson et al. (2017) have found significant positive trends in emissions in northern North America and North Eurasia over 2005−2013, contradicting previous global inversion studies based on a more limited observational network north of 50°N (Bruhwiler et al., 2014; Bergamaschi et al., 2013).

Combining atmospheric methane modelling using the CHIMERE chemistry-transport model (Menut et al., 2013) and surface observations from six continuous measurement sites, this paper aims at evaluating the information contained in methane observations concerning the type, the intensity and the seasonality of Arctic sources. The study focuses on 2012, as this is the last year for which wetland emissions are available for a set of models in a controlled framework. Section 2 describes the data and modelling tools used in this study. Section 3 analyses the simulated methane mole fractions and investigates their agreement with the observations. It also discusses the sensitivity of the model to wetland and freshwater sources, as well as to methane sinks. Section 4 concludes this study.

## 2 Data and model framework

2.1. Methane observations

Continuous methane measurements for the year 2012, from the six Arctic surface sites, have been gathered. The sites characteristics are given in Table 1, and Fig. 1 represents their position in the studied domain. Two sites are considered as remote background sites: Alert, located in North Canada, where measurements are carried out by Environment Canada (EC), and Zeppelin (Ny-Alesund), located in Svalbard archipelago on a mountaintop, and operated by the Norwegian Institute for Air Research (NILU). NOAA-Earth System Research Laboratory (NOAA-ESRL) is responsible for the measurements at Barrow observatory, which is located in northern Alaska, 8 km northeast of the city of Barrow, and at Cherskii. Cherskii and Tiksi are located close to the shores of the East Siberian Sea and the Laptev Sea, respectively. Pallas is located in northern Finland, with dominant influence from Europe. Measurements at these last two sites are carried out by the Finnish Meteorological Institute

(FMI). No data were available in Barrow in 2012 after May, due to a lapse in funding (Sweeney et al, 2016). Gaps in Cherskii (October-January), Pallas (August-mid-October), and Zeppelin (January-April) data are due to instrument issues.

Data from Alert, Barrow and Pallas were downloaded from the World Data Centre for Greenhouse Gases (WDCGG, http://ds.data.jma.go.jp/gmd/wdcgg/). Tiksi data were obtained through the NOAA-ESRL IASOA (International Arctic Systems for Observing the Atmosphere) platform (https://esrl.noaa.gov/psd/iasoa/). Zeppelin data were obtained via the InGOS (Integrated non-$CO_2$ Greenhouse Gas Observing System) project. Cherskii data were provided by NOAA. All valid data from the sites are used in this study, with no filter applied.
All data are reported in units of mole fraction, nmol mol$^{-1}$ (abbreviated ppb) on the WMO X2004 $CH_4$ mole fraction scale. Observations are available at hourly resolution at least, but in this study we make use of daily means to focus on synoptic variations, which are more appropriate for regional modelling.

2.2 Model description

The CHIMERE Eulerian chemistry-transport model (Vautard et al., 2001; Menut et al., 2013) has been used for simulations of tropospheric methane. It solves the advection-diffusion equation on a regular grid, forced using pre-computed meteorology. Our domain goes from
220 39°N to the Pole but it covers all longitudes only above 64°N, as it is not regular in terms of latitude/longitude. Its regular kilometric resolution of 35 km allows us to avoid numerical issues due to shrunken grid cells near the Pole (Berchet et al., 2016). 29 vertical levels characterize the troposphere, from the surface to 300 hPa (~9000 m), with an emphasis on the lowest layers.
The model is forced by meteorological fields from European Centre for Medium Range Weather Forecasts (ECMWF) forecasts and reanalyses (http://www.ecmwf.int). These include wind, temperature and water vapour profiles characterized by 3 h time resolution, a spatial resolution of ~0.5°, and 70 vertical levels in the troposphere. Initial and boundary
concentrations come from optimized global simulations of the LMDZ general circulation model for 2012 (Locatelli et al., 2015). These fields have a 3 h time resolution and 3.75°x1.875° spatial resolution. They are interpolated in time and space with the grid of the CHIMERE domain.

The model is run with seven distinct tracers: six correspond to the different Arctic emission sources (anthropogenic, biomass burning, geology & oceans, ESAS, wetlands, and freshwaters) and one corresponds to the boundary conditions. This framework allows us to analyse the contribution of each source in the simulated total methane mixing ratio, defined as the sum of each tracer. No chemistry is included in the standard simulations, but a sensitivity
test is made (see section 3.4).

2.3 Emission scenario

Surface emissions used here stem from a set of various inventories, models, and data-driven
studies, from which is built a reference scenario, complemented by several sensitivity scenarios. The different emission sources used are described in Table 2, along with the amount of methane emitted in the studied domain.

All types of anthropogenic emissions are provided by the EDGAR (Emission Database for
Global Atmospheric Research) v4.2 Fast Track 2010 (FT 2010) data (Olivier and Janssens-
Maenhout, 2012), which has a 0.1°x0.1° resolution. EDGAR emissions are derived from
activity statistics and emission factors. Given that the EDGARv4.2FT2010 emissions are not
available for years after 2010, the 2010 values are used for 2012 for every sector but the ones
for which FAO (Food and Agriculture Organization, http://www.fao.org/faostat/en/#data/)
and BP (http://www.bp.com/) data are available (oil and gas production, fugitive from solid,
enteric fermentation, and manure management). In this latter case, the ratio of 2012 to 2010 is
used at the country level to update the EDGAR 2010 emissions. For our domain, prior
anthropogenic emissions represent 20.5 $TgCH_4$ $yr^{-1}$, mostly from the fossil fuel industry.

Biomass burning emissions come from the Global Fire Emissions Database version 4
(GFED4.1) (van der Werf et al., 2010; Giglio et al., 2013) monthly means product. Burned
areas estimated from the MODIS spaceborne instrument are combined with the biomass
density and the combustion efficiency derived from the CASA biogeochemical model, and
with an empirically-assessed emission factor. The emissions are provided on a 0.25°x0.25°
grid. Biomass burning emissions are 3.1 $TgCH_4$ $yr^{-1}$ in our domain.

Wetland emissions in the reference scenario come from the ORCHIDEE-WET model
(Ringeval et al., 2010, 2011), which is derived from the ORCHIDEE global vegetation model
(Krinner et al., 2005). The wetland methane flux density is computed for each 0.5°x0.5° grid
cell based on the Walter et al. (2001) model. Three pathways of transport (diffusion,
ebullition and plant-mediated transport) and oxidation are included. Annual emissions from
wetlands in our domain are 29.5 $TgCH_4$ $yr^{-1}$ with the ORCHIDEE model. The version of
ORCHIDEE used in this study comes from Poulter et al. (submitted) (see also Saunois et al.
(2016)), like the ten other land surface models used for sensitivity studies (cf. section 3.2).
Following Melton et al. (2013), net methane emissions have been computed under a common
protocol; the models use the same wetland extent and climate forcings. Wetland area
dynamics are based on global wetland datasets produced with the GLWD (Global Lakes and
Wetlands Database), combined with SWAMPS (Surface WAter Microwave Product Series)
inundated soils maps. The emissions from these ten other models range from 10.1 up to
58.3 $TgCH_4$ $yr^{-1}$.

Emissions from geologic sources, including continental macro- and micro-seepages, and
marine seepages, are derived from the GLOCOS database (Etiope, 2015). They represent
4.0 $TgCH_4$ $yr^{-1}$ in our domain.

ESAS emissions are prescribed following Berchet et al. (2016), and scaled to 2 $TgCH_4$ $yr^{-1}$.
Their temporal variability is underestimated as uniform and constant emissions were applied
by emission type (hot spots and background) and period (winter/summer), based on Shakhova
et al. (2010). In particular, we assume that substantial emissions take place during the ice-
covered period through polynyas. Although a part of the emissions in ESAS can be
considered geological, all potential sources emitting in ESAS are here considered as one
distinct source.

Generally poorly or not at all represented in former atmospheric studies, freshwater emissions
were built for the purpose of this work. The inventory is based on the GLWD level 3 product
(Lehner and Döll, 2004), which provides a map of lake and wetland types at a 30 second
(~0.0083°, or 421 m x 922 m at 60°N) resolution. A total value of 15 $TgCH_4$ $yr^{-1}$ was
prescribed for freshwater emissions at latitudes above 50°N, according to several recent

studies (e.g., Walter et al. (2007): 24.5 $TgCH_4$ $yr^{-1}$ above 45°N; Bastviken et al. (2011): 13 $TgCH_4$ $yr^{-1}$ above 54°N; Wik et al. (2016): 16.5 $TgCH_4$ $yr^{-1}$ above 50°N; Saunois et al. (2016): 18 $TgCH_4$ $yr^{-1}$ above 50°N). This value was uniformly distributed over lake and reservoir grid cells, assuming that a lake or a reservoir occupies the entire grid cell. This method is simplistic, as the dependence of emissions on lake areas, depths, and types are not taken into account. The seasonality of the emissions is underestimated given that no emission takes place when the lake is frozen, and that the emission is constant after ice-out. Therefore, our inventory does not allow episodic fluxes such as spring methane bursts (Jammet et al., 2015), and emissions during ice-cover period (Walter et al., 2007). Freeze-up and ice-out dates were estimated using surface temperature data from the ECMWF ERA-Interim Reanalyses. For each lake or reservoir, freeze-up was assumed to happen after two continuous weeks below 0°C; ice-out, after three continuous weeks above 0°C. Again, this is a simplification, given that there is no simple relation between air temperature and freeze-up or ice-out (e.g., Livingston, 1999).

As a result, we built an inventory for freshwater emissions (Fig. 2a), (i) with a total budget of 9.3 $TgCH_4$ $yr^{-1}$ in our domain, consistent with the range provided by recent literature, (ii) with a regional seasonality which is similar to the one of wetland emissions, and (iii) without overlap with wetland areas, as both use the same GLWD database. The impact of this self-made inventory is also compared with the recently published work from Tan et al. (2015) for Arctic lakes (cf. section 3.3).

The more recent GLOWABO (Global Water Bodies) database (Verpoorter et al., 2014) has a higher resolution than the GLWD (0.002 vs. 0.1 $km^2$), and finds a higher combined global surface area of lakes and reservoirs (5 vs. 2.7 $10^6$ $km^2$) as it takes into account smaller lakes. By using the GLWD product for identifying both lake and wetland areas, our freshwater inventory may therefore underestimate the emitting surface area, while the wetland inventories may still include open water fluxes. Double-counting is avoided in terms of area, but not necessarily in terms of emission (Thornton et al., 2016b).

**3 Results**

3.1 Reference simulation

3.1.1 Source contributions within the domain

A simulation of seven methane tracers is run with CHIMERE for 2012. On top of methane from initial and boundary conditions, these include methane from anthropogenic sources, biomass burning, East Siberian Arctic Shelf (ESAS), geology and oceans (counting as only one source and excluding ESAS), wetlands, and freshwaters.

The boundary conditions are the dominant signal; they result from emissions coming from sources located outside of the domain, and from emissions coming from Arctic sources, which have once left the domain and then re-entered in it. The boundary condition tracer does not hold information on where the transported methane initially comes from. So, to focus on Arctic sources, the source contributions are defined here relatively to the sum of the six tracers which correspond to sources located in the domain, i.e. excluding methane resulting from the boundary conditions. The source contribution is only calculated when methane

directly coming from Arctic sources is greater than 1 ppb. One should keep in mind that this signal represents a small fraction of total atmospheric methane.

The weight of each source varies both spatially and seasonally. Figures 3 and 4 represent the mean source contributions to methane concentrations near the surface, in winter (November to May) and in summer (from June to October), respectively.

In winter, anthropogenic methane is dominant (over winter, the daily average over the domain is in the range 18-59%, with a mean of 42%). More than 80% of anthropogenic emissions come from oil, gas and coal industries. In particular, it affects western Russia (mostly due to gas production), the Khanty-Mansia region (mostly due to oil production), and south-eastern Russia (mostly due to coal mining). Oil production is also the main contributor to atmospheric
methane in continental Canada.

Geologic and oceanic emissions represent an important part of atmospheric methane in the domain, particularly in winter (11-36%, mean: 27%). Emissions from ESAS are expected to be larger in summer, when most of the area is ice-free, than in winter. However, its relative
contribution is higher in winter (8-23%, mean: 15%), when other sources, particularly from wetlands, are lower. Alaska and Northern Siberia are particularly affected by geology and ocean emissions in winter, including from ESAS.

In summer, wetland emissions are the dominant contributor (33-56%, mean: 50%) (although
anthropogenic emissions remain important in western Russia), while they are quite negligible in winter. Freshwaters too are an important contributor in summer (9-29%, mean: 19%), but of lower intensity than wetlands, except in eastern Canada and Scandinavia, where methane from lakes can exceed methane from wetlands.

Biomass burning takes place in summer (0-7%, mean: 4%), when fuel characteristics and meteorological conditions foster combustion. Although the 2012 fire emissions are particularly high (e.g., almost twice as high as the 2013 emissions) and large scale fires occur in boreal Russian and Canadian forests, their impact on methane remains limited to some regions in continental Russia.

3.1.2 Arctic source contributions at atmospheric monitoring sites

The contribution of the different sources is more quantitatively discussed in the following, focusing on the six continuous measurement sites shown in Fig. 3 and 4.

The evolution of the daily averaged source contributions at the six sites is represented in Fig. 5. In December and from January to April, methane from Arctic sources is driven by anthropogenic, ESAS and geology and oceans emissions at all sites. It is confirmed by the figures in Tables 3 and 4, which give the mean relative and absolute contributions,
respectively, for winter and summer. Over winter, anthropogenic sources account for more than 50% only in Pallas and Zeppelin. For the other four sites, anthropogenic emissions contribute between 23 and 35%, while methane from continental seepages and oceans, including ESAS, account for more than 54% of methane from Arctic sources, and up to 68% at Tiksi, corresponding to 18 ppb. ESAS emissions have the lowest impact in methane levels
in Pallas and Zeppelin (<1 ppb). Freshwaters and wetlands combined contribute between 8 and 27% in winter, corresponding to only a few ppb.

Wetland emissions start having an impact in May and dominate from June to October, fading in November (Fig. 5). Freshwater emissions present a similar seasonal cycle, except in Pallas where some contributions are seen in December-January. According to the lake inventory developed here, southernmost Scandinavian lakes have not frozen over and continue to emit until January. Elsewhere, their contribution follows the same seasonality as wetland emissions' but lagged by one month, and with a lower impact. In summer, wetland emissions are the major contributor from Arctic sources at all sites (from 48 to 70%, or from 10 to 84 ppb), and methane coming from both wetland and freshwater sources amount to at least 65% of methane coming from Arctic sources, on average, for all sites. These two major sources overshadow anthropogenic sources whose impact remains below 16%. Only Cherskii and Tiksi are substantially impacted by ESAS emissions in summer (10 and 17%, or 8 and 11 ppb, respectively). Overall, biomass burning negligibly contributes to the methane abundance at the six surface sites.

Figure 5 also shows the evolution of the simulated methane coming from Arctic sources (white line, right-hand axis). Over the year, Alert, Pallas and Zeppelin mixing ratios have lower contributions from Arctic sources (always below 60 ppb) than Barrow, Cherskii and Tiksi (sometimes more than 120 ppb). In winter, although the source repartition is different among the sites, methane levels are quite low for all of them, from 10 ppb in Alert to 26 ppb in Tiksi, on average (Table 4). However, there still are individual peaks related to either predominant anthropogenic or ESAS sources. In Alert for example, on 1$^{st}$ March, methane from Arctic sources reaches 31 ppb, 77% of which corresponds to anthropogenic sources. In Cherskii, on 5$^{th}$ April, 89% of the 45 ppb methane signal came from ESAS emissions. Contributions from geology and oceanic sources can reach the highest proportions in winter, but it repeatedly corresponds to only a few ppb of methane, up to only 14 ppb in Barrow in 4$^{th}$ December.

In summer, all measurement sites see higher methane contributions from Arctic sources, predominantly from wetland emissions, with Barrow, Cherskii, and Tiksi being more affected by them. These last three sites experience contributions greater than 45 ppb on average, while, for the three others, contributions from Arctic sources remain below 26 ppb. The freshwater signal is almost always less than the wetland signal, but even for Alert and Zeppelin, which have the lowest levels of methane coming from freshwater emissions, it sometimes exceeds 25%, with substantial corresponding contributions in ppb.

3.1.3 Comparison with observations

The simulated absolute values of total methane at the sites are shown in Fig. 6 and 7, along with the observed mixing ratios. There is good agreement between observed and simulated methane, both in terms of intensity and temporal evolution. In particular, the model shows its ability to reproduce short-term peaks and drops, which are either due to the intrusion of enriched or depleted air from outside of the domain, or directly due to the evolution of Arctic sources.

Although Arctic emissions are greater in summer, Alert, Pallas and Zeppelin have higher methane values in winter due to higher influence of air coming from lower latitudes, whose methane seasonal cycle is mostly driven by OH. Table 5 gives the differences between the mean methane in winter and the mean methane in summer for the observations and the reference simulation. The greatest seasonal cycle is seen in Pallas, the closest site to mid-latitude Europe. Tiksi is less sensitive to boundary conditions, and the influence of summer

sources produce an opposite seasonal cycle (maximum in summer), although with a weaker average amplitude than for the three sites mentioned above. Observations in Cherskii show no clear seasonal cycle, in contradiction with the simulation, particularly in September, when simulated methane from wetlands frequently exceeds 100 ppb. This discrepancy is mainly due to an overestimation of wetland emissions by ORCHIDEE in the region nearby Cherskii.

As we have seen above, these two kinds of seasonal cycle do not prevent the same kind of events from happening at the scale of a few days (synoptic variations). For instance, even if methane variability in Alert, Pallas and Zeppelin is mostly driven by the boundary conditions in winter, measurements made at these sites do hold information on Arctic (anthropogenic, geologic and oceanic) sources during particular synoptic events. And in summer, methane peaks have important contributions at all sites from wetland and freshwater emissions. Overall, with the exception of biomass burning, all sources have a substantial impact on the six measurement sites, whether it is on the scale of synoptic events of a few days or regularly occurring over the course of several months.

The overall good agreement between simulations and measurements is quantified in Table 6, which gives the mean difference between observed and simulated methane during 2012. The mean daily bias remains below 7.5 ppb for all sites, except for Cherskii, where it reaches 34.8 ppb, mostly because of a large overestimation of methane coming from wetland emissions in September. For all sites, the bias stems from an overestimation of modelled methane in summer (in the range 4.8–8.6 ppb, Cherskii excluded), which is compensated in winter by either a lower overestimation (Pallas, Tiksi, Zeppelin), or an underestimation (Alert, Barrow, Cherskii). As a result, the seasonality is well captured in Pallas, Tiksi, and Zeppelin, but is not pronounced enough in Alert (Table 5).

At Alert (Fig. 6), simulated methane is higher than the measurements in June and July. The boundary conditions may be responsible for this disagreement, given that, for several days, the measurements are lower than methane resulting from the boundary conditions alone. The absence of the methane sinks in the reference simulation may also be a reason. It may also indicate that the emissions are not well represented in the reference simulation. In August, September and October, then, the reference simulation agrees better with the measurements, although the intensity of some modelled peaks may be too low.

The results of our reference simulation depend on the hypotheses made, especially on source distribution (cf. Fig.S1-S6) and absence of methane sinks. The impact of wetland and freshwater source distribution and of methane sinks on modelled atmospheric methane is investigated in the next sections as sensitivity tests.

3.2 Impact of different wetland emission models

As noted previously, wetland emissions represent the main source of methane in the Arctic, explaining at least 48% of the methane signal coming from Arctic sources for all six measurement sites in summer on average. Therefore, the representation of wetland emissions in Arctic methane modelling is crucial. This is why the outputs of ten other land surface models than ORCHIDEE have been tested, for June to October 2012 (assuming significant wetland emissions only take place at this time of year). The impact of the different land surface models is assessed focusing on the four sites that provide data uniformly distributed along these five months (Alert, Cherskii, Tiksi and Zeppelin).

The eleven land surface models are described in Poulter at al. (submitted) and Saunois et al. (2016). Wetland emissions are mostly located in Scandinavia, between the Ob and Yenisei rivers and between the Kolyma and Indigirka rivers in Russia, Nunavut (NU) and Northwest Territories (NT) in Canada, and in Alaska, with large discrepancies among the models even if they use the same wetland emitting zones (cf. section 2.3). Emissions from all models and their evolution over the year is illustrated in Fig. S5 and S6. For all models, emissions start in May and end in October. The maximum in emission is reached in June (for the LPJ-wsl, CTEM, and DLEM models) or in July. Only the LPX-Bern and SDGVM models have maximum emissions in August and September, respectively. The latter has the highest emissions of all models in September and October, due to its ~2-month shifted seasonality, but its emissions in November are close to zero, like the other models. The emission intensities vary from one model to another (Table 2). Three models have emissions below 20 $TgCH_4$, four below 30 $TgCH_4$, three below 40 $TgCH_4$; LPJ-MPI stands apart with 58.3 $TgCH_4$. Overall, ORCHIDEE stands in the middle of the models range.

Given the sensitivity to the variability of methane coming from the boundary conditions in Alert and Zeppelin, and its likely overestimation in June-July (see section 3.1.3), the bias alone is not a good criterion for evaluating the different wetland models. Instead, Figure 8 shows Taylor diagrams of the comparisons between methane simulated with the outputs of eleven different land surface models and the measurements. At Alert, SDGVM is the best performing model in terms of its correlation with the measurements (correlation coefficient R of 0.85), and one of the best in terms of its standard deviation (8.9 vs. 11.3 ppb for the measurements). In Zeppelin, SDGVM has again the best correlation coefficient (R=0.87). Given its shifted seasonality compared to the other models, SDGVM produce the lowest methane values in June and partly in July, i.e. the best agreement with the measurements, both in Alert and Zeppelin. In September and October, when the reference simulation can be too low, the simulation with SDGVM is one of the highest, performing well at capturing some methane peaks. Although it has the third and second worst biases in Alert and Zeppelin, respectively, these biases are the least variable over the 5-month period (Table 7). As a result, it seems to be the most convincing wetland model regarding the comparisons at Alert and Zeppelin.

In Tiksi, the high variability and high values of methane peaks lead to low correlation coefficients, as the model is not fully able to reproduce the short term variability whatever the wetland emission. However, SDGVM reaches a correlation coefficient of 0.60. SDGVM and ORCHIDEE have standard deviations similar to the measurements and two of the three lowest biases. However, ORCHIDEE's correlation coefficient is only 0.39.

In Cherskii, like in Tiksi, the model has troubles reproducing the variability of the measurements, and this can lead to high biases. However, CLM4.5 and LPX-Bern have biases below 9 ppb and correlation coefficients above 0.62, with similar standard deviations. It is worth noting that SDGVM and ORCHIDEE have here the two worst correlation coefficients. Again, the simulation with ORCHIDEE has unexpectedly extreme values in September, up to 2925 ppb, certainly due to outlying high emissions in the Kolyma and Indigirka region in this month. Indeed, according to ORCHIDEE, 1.4 $TgCH_4$ is emitted in this region (65°N-73°N, 140°E-170°E) for September alone, while the median model emits only 0.1 $TgCH_4$.

The comparison between the measurements and the simulations performed with the outputs of ten different land surface models and with the reference scenario, show that no wetland emission model performs perfectly. SDGVM and LPX-Bern, which is overall the least biased

model, seem to be the two most reliable models on average. These models are characterized by low emissions in early summer/late spring. ORCHIDEE, except in Cherskii, has a fair average performance, compared to the other models. On the contrary, LPJ-MPI is a clear outlier, leading to methane values that are too high.

The results obtained in section 3.1 appear to be sensitive to the choice of the land surface model. More effort is needed to better represent the location, timing and magnitude of Arctic wetland emitting zones (Tan et al., 2016). Continuous observations clearly offer a good constraint to handle this challenge.

3.3 Impact of the bLake4Me freshwater emission model

Freshwater emissions are the second main contributing source in the Arctic in summer, explaining between 11% and 26% of the atmospheric signal at the six measurement sites on average. As was previously noted, there is a large uncertainty affecting the distribution and magnitude of this particular source. This is why an alternative lake emission inventory is tested here. bLake4Me is a one-dimensional, process-based, climate sensitive lake biogeochemical model (Tan et al., 2015; Tan and Zhuang, 2015a,b). Model output used here corresponds to the 2005-2009 average.

The difference between the inventory used in the reference simulation and the one based on bLake4Me is shown in Fig. 2b. Since bLake4Me's output is only available above 60°N, the reference simulation's inventory is used between the edges of the domain and 60°N, therefore showing no difference in this area. The total freshwater emission with bLake4Me is 13.6 $TgCH_4$ $yr^{-1}$, i.e. 4.3 $TgCH_4$ $yr^{-1}$ more than in the reference simulation. The difference mostly takes place between the Kolyma and Indigirka rivers, where bLake4Me's emissions happen all year, in the centre of the Khanty-Mansia region, and in the Northwest Territories in Canada. On the contrary, emissions in Scandinavia and northwestern Russia are lower by about 1 $TgCH_4$ $yr^{-1}$ in bLake4Me. Both inventories have their maximum emission in August.

Figure 9 represents the difference between the absolute value of the bias calculated with the simulation using the bLake4Me inventory and the absolute value of the bias of the reference simulation. A positive value (black dots), therefore, means that the freshwater inventory developed for the reference simulation performs better than the bLake4Me inventory. For Alert, Barrow, Pallas, and Zeppelin, differences in the bias generally remain within ±10 ppb. The largest change in methane levels brought by the variant lake emission scenario is seen in Cherskii, where simulated methane is higher all year long, with differences of more than 100 ppb in December-February (Fig. S8). These winter emissions from ice-covered lakes in the bLake4Me inventory are triggered by intense point-source ebullition from the thermokarst margins of yedoma lakes (Tan et al., 2015). In Cherskii, the bLake4Me inventory does not improve the simulation, given that the reference simulation already overestimates methane in summer, and underestimates the measurements by only a few ppb in winter. The increased bias in winter may be caused by an overestimation of the lake edge effect in bLake4Me. In Tiksi, simulated methane is higher all year long too, but the difference with the reference simulation never exceeds 50 ppb. The simulation is not improved with this inventory at Tiksi. The bias over the year (Table 6), which already showed an overestimation of the reference simulation, is now twice as large with the variant inventory. In Barrow, more than 100 additional ppb in methane coming from lakes happen in July-August, but no data are available to assess their validity. In the other months, the effect of the variant lake emissions is negligible.

In Alert and Zeppelin, using bLake4Me inventory increases simulated methane by a few ppb in July-September, with no major changes during the rest of the year. This leads to an increase in the bias, although this can also improve agreement with the measurements for some periods, particularly in September, when the reference simulation underestimates some methane peaks. Table 5 shows that the changes brought by the new inventory worsen the seasonality simulated at these two stations.

Only in Pallas does the bLake4Me inventory lead to lower simulated methane, particularly in winter, linked to the shortened season of freshwater emissions in Scandinavia. As a consequence, the bias is improved from -5.3 to -4.9 ppb over the year (Table 6).

Although bLake4me produces physical outputs of freshwater emissions, and is therefore far more advanced than the crude inventory developed here for the reference simulation, no significant improvement is found in comparisons between simulated and observed methane at the six measurement sites. Once again, as stated for wetlands (section 3.2), the distribution and magnitude of lake emissions can be critical to correctly reproducing methane concentrations at sites located nearby (e.g., Cherskii). Using such observational stations combined with a chemistry-transport model offers a good constraint to improve the magnitude and location of methane emissions from lakes in the Arctic.

3.4 Impact of the methane sinks

Regional modelling of atmospheric methane generally does not consider methane sinks, focusing more on synoptic variations than on long-term changes. This is justified by the rather long methane lifetime (~9 years) regarding the synoptic to seasonal time scales. However, even if air masses are expected to stay in the Arctic domain (as defined here) up to only a few weeks, the cumulated impact of the different sinks on the concentrations might not be negligible and should at least be quantified.

The main atmospheric loss of methane results from OH oxidation in the troposphere. OH concentrations are higher in summer and above continents, as its production is controlled by solar radiation, albedo, and the concentrations of $NO_x$ and $O_3$. In the Arctic, OH thus reaches its lowest values in winter (below $0.5 \times 10^5$ molec. $cm^{-3}$, mass-weighted), and is at its maximum in July ($11$-$12 \times 10^5$ molec. $cm^{-3}$). OH daily data coming from the TransCom experiment (Patra et al., 2011; Spivakovsky et al., 2000) were included in CHIMERE as prescribed fields and the JPL recommended reaction rate constant $k_{OH+CH4} = 2.45 \times 10^{-12} \times exp^{-1775/T}$ (Burkholder et al., 2015) was used.

Figure 10a shows the difference between the reference simulation and the simulation including methane oxidation by OH, thus representing the effect of the methane sink due to OH on the mixing ratios (set to a positive value). As expected, the impact is mostly visible in summer. Even if the general pattern is similar among the sites – a progressive increase in the OH sink effect from March to July, when it can be as high as 12 ppb, and a symmetric decrease until November –, the daily variability in the OH sink effect is not the same for all sites. Pallas, for example, has the strongest variability. This variability stems from the disparity in the proximity/distance of the origin of the air masses observed at the sites, combined with the heterogeneity in the distribution of OH concentrations.

The second potential chemical sink lies in the oxidation of methane by chlorine (Cl) in the marine boundary layer. Theoretical prescribed Cl fields were thus included in CHIMERE, following the recommended scenario described in Allan et al. (2007). Cl atoms are concentrated in the marine boundary layer, above ice-free zones. Daily sea ice data from the EUMETSAT Ocean and Sea Ice Satellite Application Facility (OSI SAF, http://osisaf.met.no/p/ice/) were applied to define the location of Cl non-zero concentrations. The seasonal evolution of Cl concentrations makes them close to zero in December-January and maximum in July-August ($17\text{-}18 \times 10^3$ molec. cm$^{-3}$). The reaction rate constant $k_{Cl+CH4} = 7.1 \times 10^{-12} \times exp^{-1270/T}$ (Burkholder et al., 2015) was used. As it can be seen in Fig. 10b, the impact of this sink on atmospheric methane signal is negligible and remains below 1 ppb.

Uptake of methane from methanotrophic soil bacteria is considered here as a surface sink. We use here the monthly 1°x1° climatology by Ridgwell et al. (1999). Depending on the soil water content and temperature, this sink is effective between March and October, with a maximum in August. Over the year, its intensity amounts to 3.1 TgCH$_4$ yr$^{-1}$. The impact of this sink is plotted in Fig. 10c and remains below 2 ppb for Alert and Zeppelin and not much more for Pallas and Barrow. The impact is more important for Cherskii and Tiksi, where it reaches about 10 ppb in late September. However we have not considered the more detailed soil uptake of Zhuang et al. (2013) and high affinity methanotrophic consumption as described in Oh et al. (2016), which might lead to undervalue our estimation of this effect.

We finally investigate whether the integration of these three methane sinks improves the fit to observed methane mixing ratios. Figure 11 shows simulated methane at Alert, including the cumulated effects of the three sinks, and compares it to the reference simulation and to the measurements. Indeed, for all sites, the reference simulation is too high in summer, but in Alert in particular, it does not reproduce properly the sharp decrease in methane happening from April to July (~40 ppb). The addition of the sinks helps fill the gap with the measurements. Biases in summer in Alert, Pallas, Tiksi and Zeppelin are in the range 0.2–3.0 ppb, whereas they are 4.8–8.6 ppb in the reference simulation. Table 6 gives the yearly biases including the effect of the sinks, showing a positive effect for all sites (except Barrow). However, their effect on the seasonal amplitude is not homogeneous (Table 5). The sinks make the seasonal cycle more marked in Alert, Pallas and Zeppelin. However, for these last two sites, as the simulated methane is too high in winter, the amplitude becomes excessive. In Tiksi, where the seasonal cycle is opposite, the sinks tend to lessen it.

On average, including the sink processes, and especially OH chemistry, appears important to better simulate methane. However, as expected, these loss processes are not sufficient to fully explain the discrepancies in the seasonal variations between the model and the measurements.

## 4 Conclusion

Atmospheric methane simulations in the Arctic have been made for 2012 with a polar version of the CHIMERE chemistry-transport model, implemented with a regular $35 \times 35$ km resolution. All known major anthropogenic and natural sources have been included and correspond to individual tracers in the simulation, in order to analyse the contribution of each one of them. In winter, the Arctic is dominated by anthropogenic emissions. Emissions from continental seepage and oceans, including from the ESAS, also play a decisive part in more

limited parts of the region. In summer, emissions from wetland and freshwater sources dominate across the entire region.

The simulations have been compared to six continuous measurement sites. Half of these sites have their seasonality mainly driven by air from outside of the Arctic domain studied here, with higher concentrations in winter than in summer, although Arctic sources are stronger in summer. The model is globally able to reproduce the seasonality and magnitude of methane concentrations measured at the sites. All sites are substantially impacted by all Arctic sources, except for biomass burning. In winter, when methane emitted by Arctic sources is lower, the sites are more sensitive to either anthropogenic or ESAS emissions on the scale of a few days; during the whole summer, they are more sensitive to wetland and freshwater emissions.

The main disagreement between the simulated and observed methane mixing ratios may stem from, in part, inaccurate boundary conditions, overestimation or mis-location of some of the sources, particularly during the May-July time period, or lack of methane sinks. We have conducted a series of sensitivity tests, varying wetland emissions, freshwater emissions, and including methane sinks.

On top of the wetland emissions computed by the land surface model ORCHIDEE (used in our reference simulation), the outputs of ten other process-based land surface models have been tested. Among them, the SDGVM and LPX-Bern models appear to be the most convincing at reconciling the simulations with the measurements. These models have lower emissions than most of the models in May-July, and reach a maximum of emission later, in September and August, respectively, while the others have their maximum in June-July. Over the wetland emission season, they both have lower emissions than ORCHIDEE (19 and 26 vs. 30 $TgCH_4$ $yr^{-1}$). These results suggest a seasonality of wetland emissions shifted towards autumn, which is supported by Zona et al. (2016). The forward modelling study of Warwick et al. (2016) also reached the same conclusions. To better capture the seasonal cycle of methane, wetland emissions needed to start no sooner than June and peak between July and September. This result was backed by isotopologues data that suggested large contributions from a biogenic source until October. In subsequent modelling studies, if wetland emission models still have the same seasonality, ways to somehow force winter emissions should be considered. On the contrary, our results do not support a scenario of large early emissions due to a spring thawing effect, as proposed by Song et al. (2012), although they do not exclude episodic fluxes during spring thaw (Jammet et al., 2015). Geographic distribution is also important. In particular, ORCHIDEE overestimates methane at Cherskii and Tiksi in September, probably due to over estimating emissions in the nearby Kolyma region.

The influence of freshwater emissions, which account for 11−26% of the methane signal from Arctic sources in summer at the six sites, is also assessed, and found to be significant. Our simple inventory, where a prescribed total budget of 9.3 $TgCH_4$ $yr^{-1}$ is uniformly distributed among all lakes and reservoirs in our domain, is compared to the 13.6 $TgCH_4$ $yr^{-1}$ emission derived from the bLake4Me process-based model. Overall, the latter overestimates methane at the six sites and does not bring a clear improvement to simulated methane within our modelling framework.

The inclusion of the major methane sinks (reaction with OH and soil uptake) in regional methane modelling in the Arctic is shown to improve the agreement with the observations. The cumulated impact of the sinks significantly decreases bias in the simulations at the sites. Reaction with Cl in the marine boundary layer, on the contrary, has a negligible impact.

Our work shows that an appropriate modelling framework combined with continuous observations of atmospheric methane enables us to gain knowledge on regional methane sources, including those which are usually poorly represented such as freshwater emissions. Further understanding and knowledge of the Artic sources may be obtained by combining tracers other than methane, such as methane isotopologues, within forward or inverse atmospheric studies. Such a study would gain in robustness with a wider and more representative atmospheric observational network. It is therefore of primary interest, considering the changing climate and the high climate sensitivity of the Arctic region, to maintain and further develop methane atmospheric observations at high latitudes, considering both remote and in-situ observations. So far, remote sensing of atmospheric methane is mainly based on sunlight absorption, thus not appropriate during high latitude winter. After 2020, the MERLIN space mission, based on a LIDAR technique, should bring an interesting complement to the surface and actual remote sensing observations (Kiemle et al., 2014), though with lower time resolution than continuous surface stations.

*Acknowledgments.* We thank the principal investigators of the observation sites which were used in this study for maintaining methane measurements at high latitudes and sharing their data. This work has been supported by the Franco-Swedish IZOMET-FS "Distinguishing Arctic $CH_4$ sources to the atmosphere using inverse analysis of high frequency $CH_4$, $^{13}CH_4$ and $CH_3D$ measurements" project. The study extensively relies on the meteorological data provided by the ECMWF. Calculations were performed using the computing resources of LSCE, maintained by F. Marabelle and the LSCE IT team.

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

Table 1. Description of the six continuous measurement sites used in this study.

| Sites | Coordinates | Altitude a.s.l. / Intake height a.g.l. (m) | Nb of hourly data in 2012 | Operator | References |
|---|---|---|---|---|---|
| Alert | 82.45°N, 62.51°W | **185 / 10** | 6769 | Environment Canada | Worthy et al. (2013) |
| Barrow | 71.32°N, 156.60°W | 11 / 16 | 1752 | NOAA-ESRL | Dlugokencky et al. (1995) |
| Cherskii | 68.61°N, 161.34°E | 31 / 3-34 | 4642 | NOAA-ESRL | |
| Pallas | 67.97°N, 24.12°E | 560 / 7 | 5078 | Finnish Meteorological Institute | Aalto et al. (2007) |
| Tiksi | 71.59°N, 128.92°E | **19 / 10** | 7957 | Finnish Meteorological Institute | Uttal et al. (2013) |
| Zeppelin | 78.91°N, 11.89°E | 475 / 15 | 5969 | NILU | Myhre et al. (2014) |

Table 2. Methane emissions in the studied polar domain, for the reference simulation, and for other scenarios. Total emissions for the reference scenario amount to 68.5 TgCH$_4$.

| Type of source | Reference scenario | Emissions (TgCH$_4$) | Variant scenarios | Emissions (TgCH$_4$) |
|---|---|---|---|---|
| Anthropogenic | Based on Edgar 2010. Olivier and Janssens-Maenhout (2012) | 20.5 | - | - |
| Biomass burning | GFED4.1. van der Werf et al. (2010) | 3.1 | - | - |
| Geology and oceans | Based on Etiope (2015) | 4.0 | - | - |
| ESAS | Based on Berchet et al. (2016) | 2.0 | - | - |
| Wetlands | ORCHIDEE land surface model. (Ringeval et al., 2010, 2011; S. Peng, private comm.) | 29.5 | 10 models from Poulter et al. (submitted) | 10.1–58.3 |
| | | | CLM4.5[1] | 31.0 |
| | | | CTEM[2] | 25.2 |
| | | | DLEM[3] | 21.8 |
| | | | JULES[4] | 38.3 |
| | | | LPJ-MPI[5] | 58.3 |
| | | | LPJ-wsl[6] | 10.1 |
| | | | LPX-Bern[7] | 19.4 |
| | | | SDGVM[8] | 26.2 |
| | | | TRIPLEX-GHG[9] | 15.4 |
| | | | VISIT[10] | 30.0 |
| Freshwaters | Our inventory, based on the GLWD lakes location map, Lehner and Döll (2004) | 9.3 | Based on bLake4Me, Tan et al. (2015) | 13.6 |

[1] Riley et al. (2011), Xu et al. (2016). [2] Melton and Arora (2016). [3] Tian et al. (2010, 2015). [4] Hayman et al. (2014). [5] Kleinen et al. (2012). [6] Hodson et al. (2011). [7] Spahni et al. (2011). [8] Woodward and Lomas (2004), Cao et al. (1996). [9] Zhu et al. (2014, 2015). [10] Ito and Inatomi (2012).

**Table 3.** Mean source contributions (in %) to atmospheric $CH_4$ (excluding $CH_4$ resulting from the boundary conditions) simulated by CHIMERE at the six observation sites, over winter (November-May, left value) and summer (June-October, right value) 2012. In bold font the major source at each site is highlighted for both seasons.

| | Mean source contribution (winter / summer) (%) | | | | | |
| | Anthropogenic | Biomass burning | Geology & oceans | ESAS | Wetlands | Freshwaters |
|---|---|---|---|---|---|---|
| Alert | 35 / 7 | 0 / 2 | **37** / 14 | 17 / 7 | 7 / **48** | 4 / 21 |
| Barrow | 25 / 4 | 0 / 1 | **40** / 10 | 25 / 6 | 7 / **53** | 4 / 24 |
| Cherskii | 23 / 3 | 0 / 1 | 24 / 3 | **41** / 11 | 9 / **70** | 2 / 12 |
| Pallas | **56** / 11 | 0 / 1 | 12 / 4 | 5 / 2 | 10 / **56** | 17 / 26 |
| Tiksi | 25 / 6 | 0 / 2 | 24 / 7 | **44** / 17 | 6 / **57** | 2 / 11 |
| Zeppelin | **53** / 16 | 0 / 2 | 22 / 11 | 14 / 7 | 7 / **48** | 4 / 17 |

**Table 4.** Same as Table 3, but for the absolute values, in ppb.

| | Mean source contribution (winter / summer) (ppb) | | | | | | |
| | Anthropogenic | Biomass burning | Geology & oceans | ESAS | Wetlands | Freshwaters | Total |
|---|---|---|---|---|---|---|---|
| Alert | **4** / 2 | 0 / 1 | 3 / 2 | 2 / 2 | 1 / **11** | 0 / 4 | 10 / 22 |
| Barrow | 4 / 1 | 0 / 1 | **5** / 4 | 5 / 2 | 1 / **26** | 1 / 12 | 16./ 45 |
| Cherskii | 4 / 2 | 0 / 1 | 3 / 2 | **11** / 8 | 2 / **84** | 0 / 10 | 21 / 107 |
| Pallas | **7** / 3 | 0 / 0 | 1 / 1 | 0 / 1 | 1 /**15** | 2 / 7 | 11 / 26 |
| Tiksi | 6 / 3 | 0 / 1 | 5 / 3 | **13** / 11 | 2 / **36** | 0 / 7 | 26 / 61 |
| Zeppelin | **6** / 3 | 0 / 0 | 2 / 2 | 1 / 2 | 1 / **10** | 0 / 3 | 10 / 21 |

**Table 5.** Difference between the means of $CH_4$ calculated during winter (November-May 2012) and summer (June-October 2012). Calculations are made only for days when measurements are available. No data are available in Barrow after May.

| | Winter – Summer difference (ppb) | | | | |
| | Measurements | Reference simulation | Simulation w/ bLake4Me | Reference simulation w/ sinks | Number of available days in winter/summer |
|---|---|---|---|---|---|
| Alert | 23 | 11 | 10 | 16 | 168 / 148 |
| Cherskii | 0 | -83 | -73 | -75 | 102 / 106 |
| Pallas | 25 | 26 | 22 | 31 | 203 / 68 |
| Tiksi | -5 | -7 | -10 | 0 | 207 / 136 |
| Zeppelin | 16 | 15 | 13 | 19 | 103 / 149 |

**Table 6.** Mean difference (and standard deviation) between observed and simulated $CH_4$ (in ppb), calculated on a daily basis, at six continuous measurement sites.

| | Bias (std) (ppb) | | | |
| --- | --- | --- | --- | --- |
| | Reference simulation | Simulation w/ bLake4Me | Reference simulation w/ sinks | Nb of days |
| Alert | -2.2 (11.0) | -3.8 (11.7) | 0.8 (8.7) | 308 |
| Barrow | 7.5 (12.5) | 5.3 (13.1) | 8.0 (10.0) | 136 |
| Cherskii | -34.8 (104.1) | -60.9 (111.4) | -30.4 (103.0) | 208 |
| Pallas | -5.3 (17.2) | -4.9 (15.9) | -3.6 (17.4) | 257 |
| Tiksi | -5.3 (20.2) | -12.8 (20.5) | -2.7 (20.7) | 329 |
| Zeppelin | -4.1 (10.4) | -5.3 (10.6) | -0.8 (9.3) | 252 |

**Table 7.** Mean difference (and standard deviation) between observed and simulated $CH_4$ (in ppb), calculated on a daily basis between June and October, at four continuous measurement sites, for eleven land surface models.

| | ORCHIDEE | CLM4.5 | CTEM | DLEM | JULES | LPJ-MPI | LPJ-wsl | LPX-Bern | SDGVM | TRIPLEX-GHG | VISIT | Nb of days |
| --- | --- | --- | --- | --- | --- | --- | --- | --- | --- | --- | --- | --- |
| Alert | -6.9 (10.6) | -7.8 (11.8) | -3.6 (10.7) | -10.1 (15.4) | -6.8 (9.9) | -21.9 (19.7) | 0.4 (10.7) | -2.5 (9.4) | -7.9 (6.0) | -1.6 (10.1) | -5.0 (12.4) | 146 |
| Cherskii | -67.5 (133.5) | -0.5 (20.8) | 10.0 (19.2) | -12.4 (21.4) | 14.2 (20.9) | -125.8 (75.0) | 18.3 (21.1) | -7.3 (22.2) | -12.2 (43.1) | 21.5 (20.1) | -5.8 (23.5) | 105 |
| Tiksi | 3.6 (27.1) | 8.0 (28.3) | 23.4 (22.5) | 4.6 (27.9) | 24.7 (24.7) | -48.5 (63.3) | 33.1 (24.9) | 16.4 (22.9) | 4.9 (21.9) | 30.6 (24.6) | 16.9 (28.1) | 134 |
| Zeppelin | -3.3 (11.2) | -4.5 (11.8) | -1.5 (11.3) | -4.2 (13.1) | -4.4 (10.2) | -16.4 (18.1) | 3.1 (11.9) | -0.7 (10.4) | -4.9 (9.6) | 1.1 (11.1) | -1.1 (13.2) | 147 |

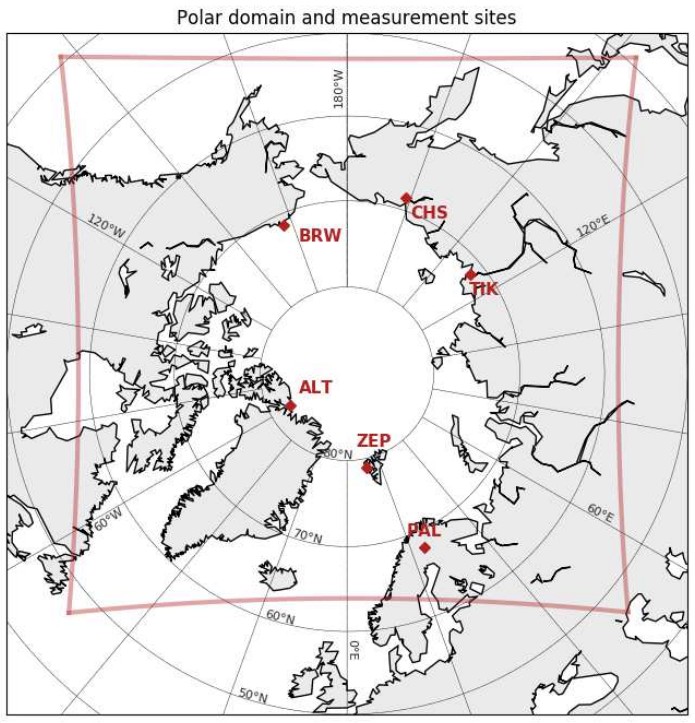

**Figure 1.** Delimitation of the studied Polar domain and location of the six continuous measurement sites used in this study. ALT: Alert. BRW: Barrow. CHS: Cherskii. PAL: Pallas. TIK: Tiksi. ZEP: Zeppelin.

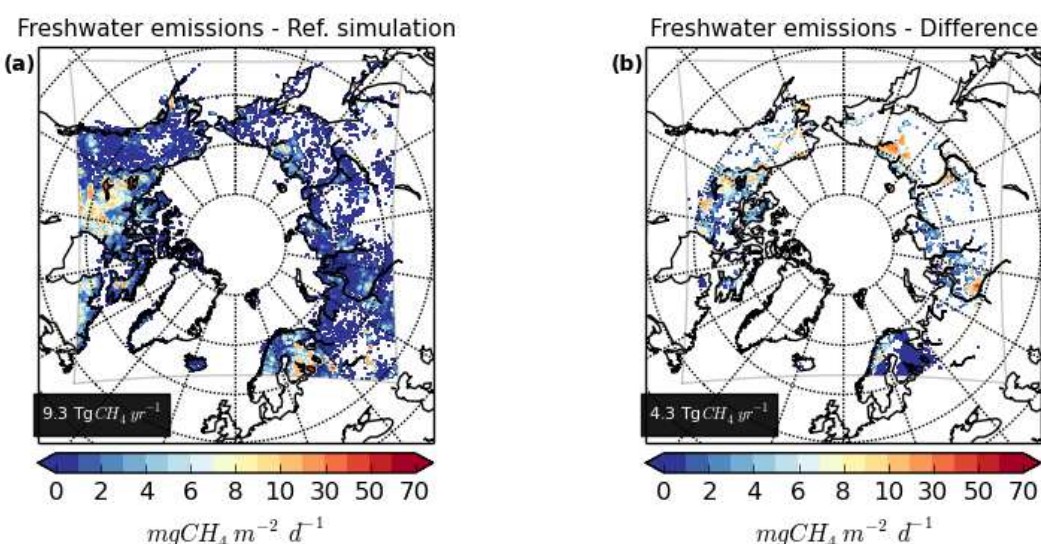

**Figure 2.** (a) Freshwater methane emissions used in the reference simulation. (b) Difference between the inventory based on the bLake4Me lake emission model (Tan and Zhuang, 2015) and the one used in the reference simulation. For both maps, blank areas in the domain correspond to zero emission.

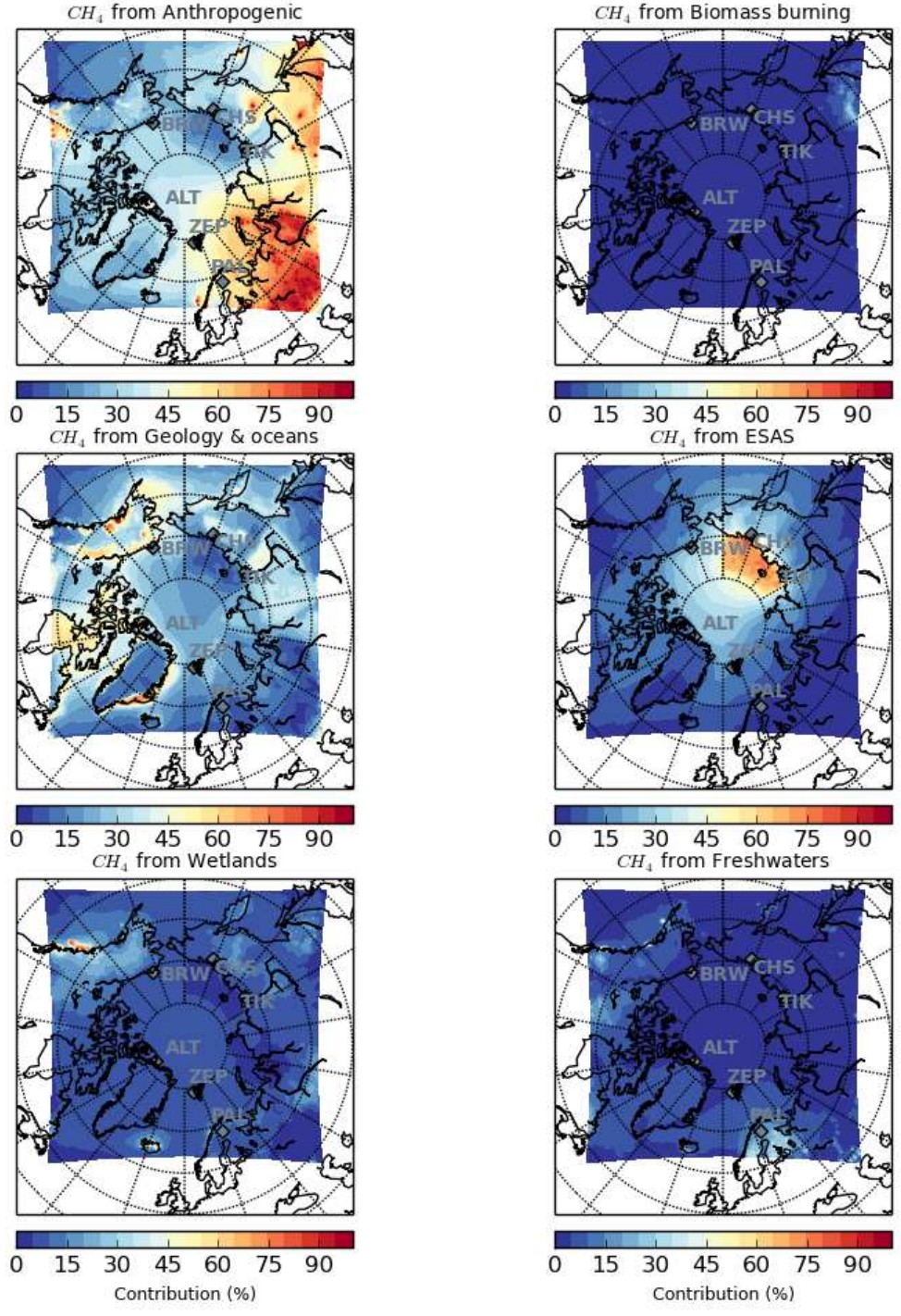

**Figure 3.** Mean sources contributions (in %) to the CH₄ abundance (excluding CH₄ resulting from the boundary conditions) simulated by CHIMERE at 990 hPa, over November-December and January-May 2012.

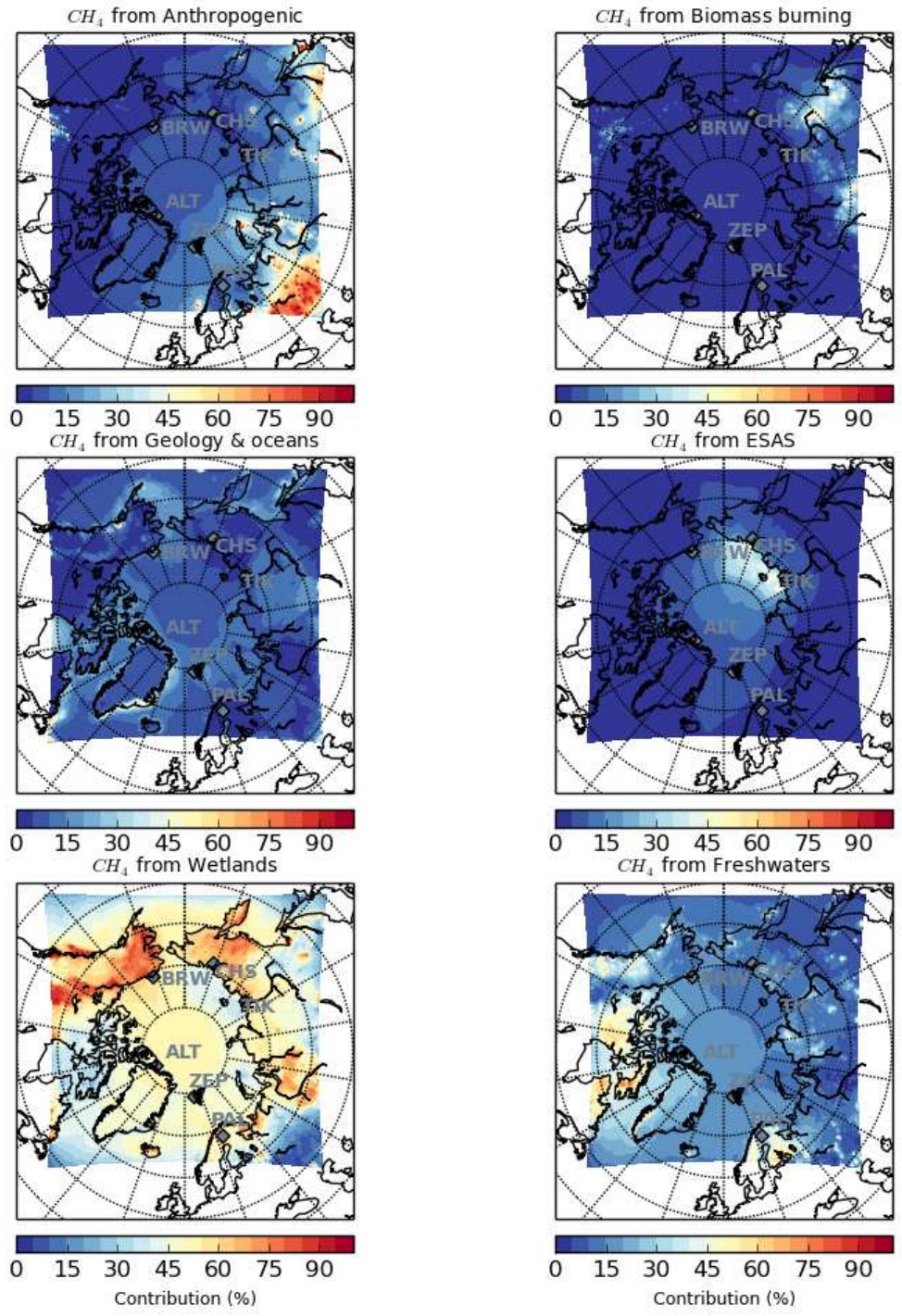

**Figure 4.** Mean sources contributions (in %) to the CH₄ abundance (excluding CH₄ resulting from the boundary conditions) simulated by CHIMERE, at 990 hPa, over June-October 2012.

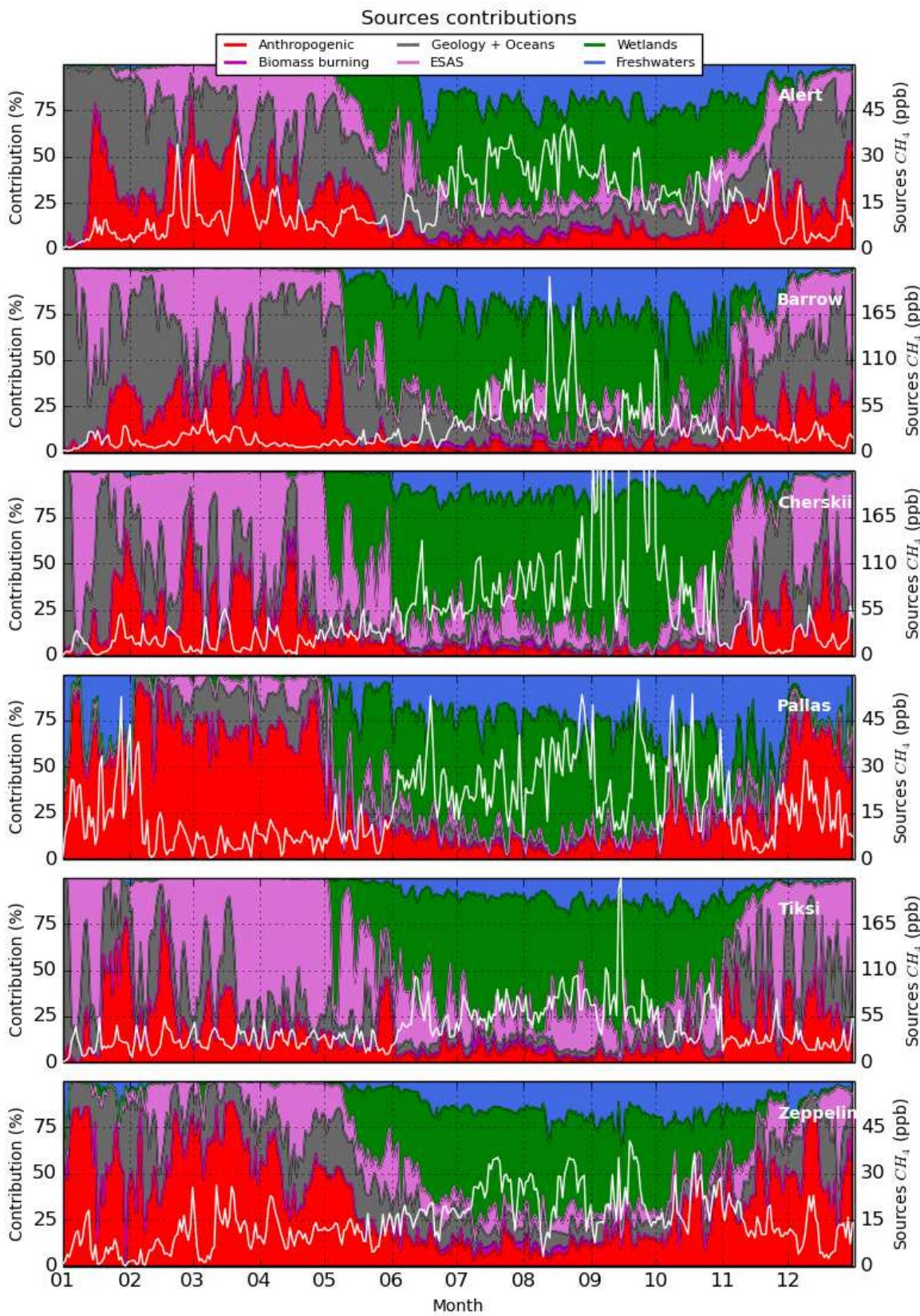

**Figure 5.** Sources contributions (in %, left-hand axis) to the CH$_4$ abundance (excluding CH$_4$ resulting from the boundary conditions) simulated by CHIMERE, at six measurement sites, in 2012. Red: anthropogenic emissions. Magenta: biomass burning. Grey: geology and oceans. Pink: ESAS. Green: wetlands. Blue: freshwaters. The white line represents the CH$_4$ mixing ratio resulting from all the sources emitted in the domain (in ppb, right-hand axis). Maximum contribution for Cherskii CH$_4$ exceeds the chosen scale and reaches 1021 ppb.

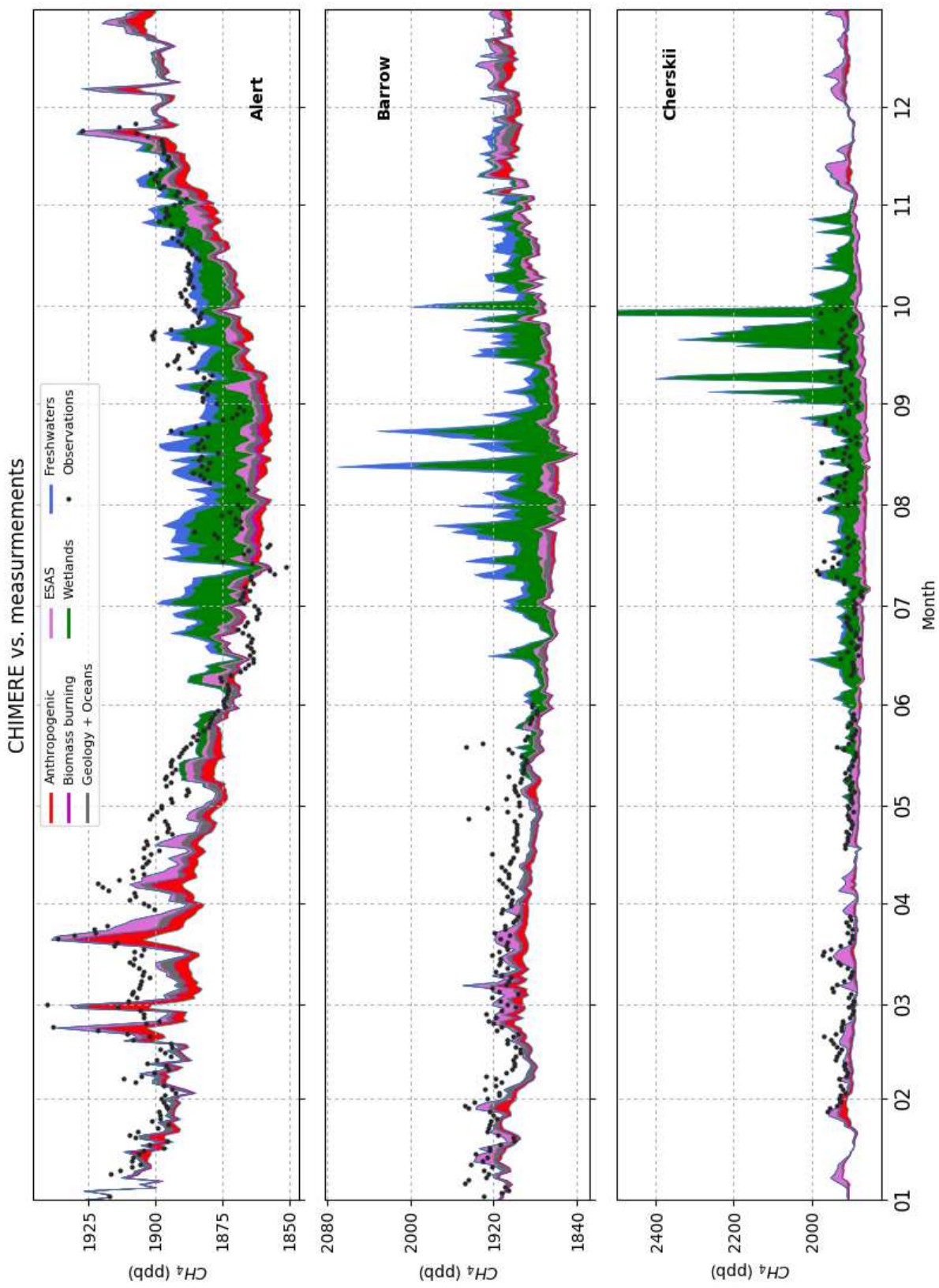

**Figure 6.** Time series of simulated (in colour) and observed (black points) methane mixing ratios in ppb, at Alert, Barrow and Cherskii, in 2012. The baseline is the contribution of the boundary conditions alone. Time resolution for simulations and observations is 1 day. Maximum for Cherskii CH₄ exceeds the chosen scale limit and reaches 2925 ppb.

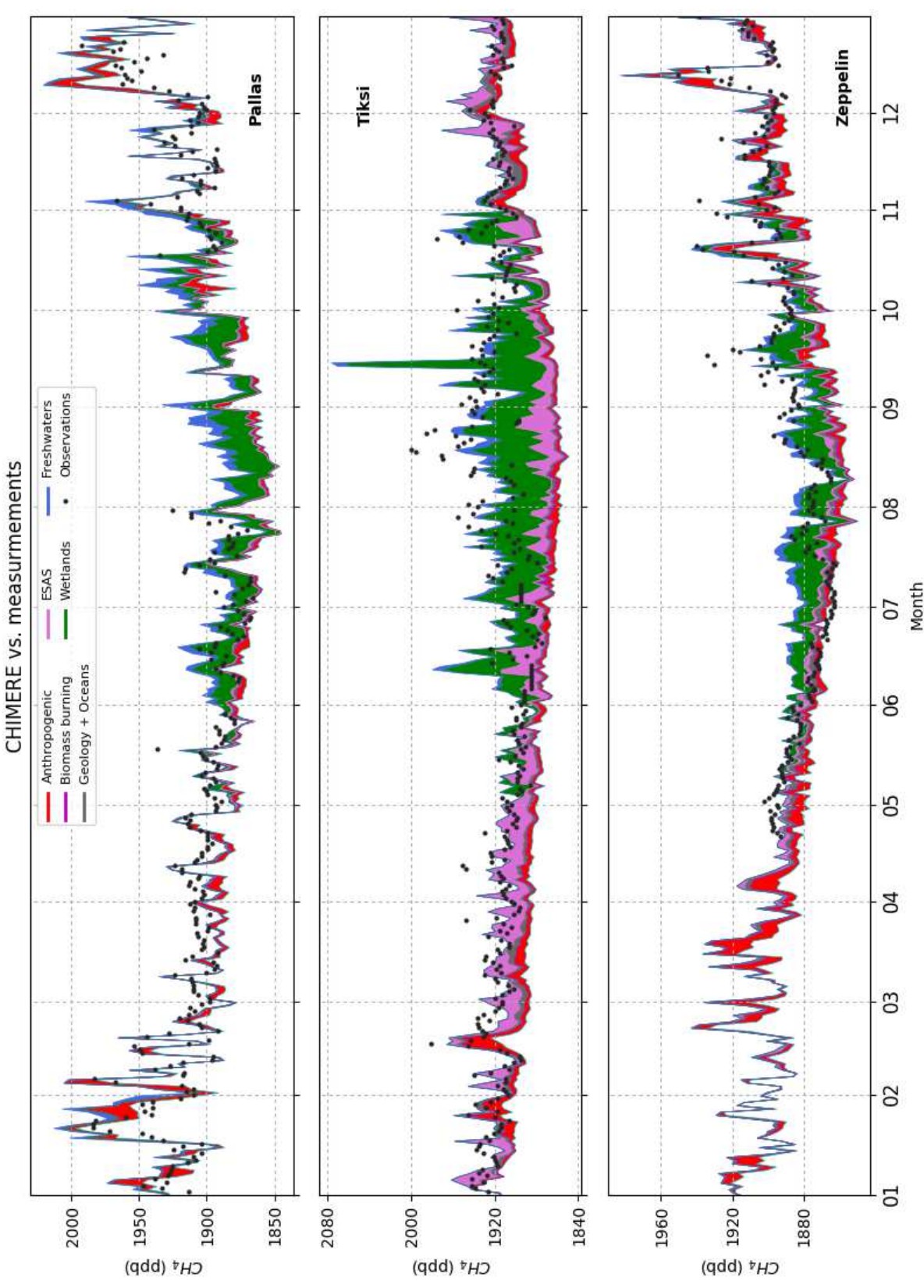

**Figure 7.** Same as Fig. 6, for Pallas, Tiksi and Zeppelin.

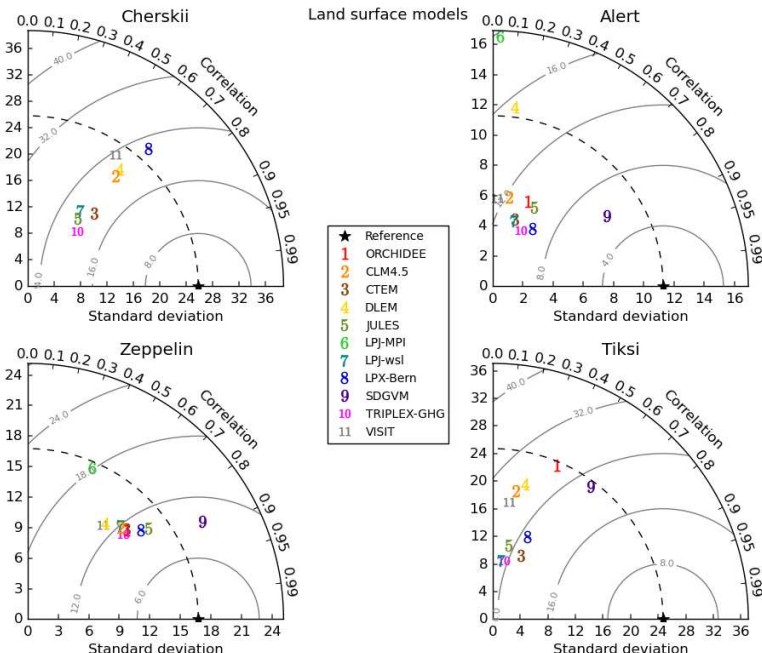

**Figure 8.** Taylor diagram representations of the comparison between observations (star marker) and CH$_4$ simulations using the outputs of 11 land surface models, at four measurement sites (Cherskii, Alert, Zeppelin and Tiksi). If we consider model 6 in Zeppelin: its correlation with observations is related to the azimuthal angle (R=0.4); the centred root-mean square (RMS) difference between simulated and observed CH$_4$ is proportional to the distance from the star marker on the x-axis, indicated by the grey contours (RMS=18 ppb); the standard deviation of simulated CH$_4$ is proportional to the radial distance from the origin (std=16 ppb). ORCHIDEE, LPJ-MPI and SDGVM, and LPJ-MPI alone do not appear in the Cherskii and Tiksi plots, respectively, because of higher standard deviations.

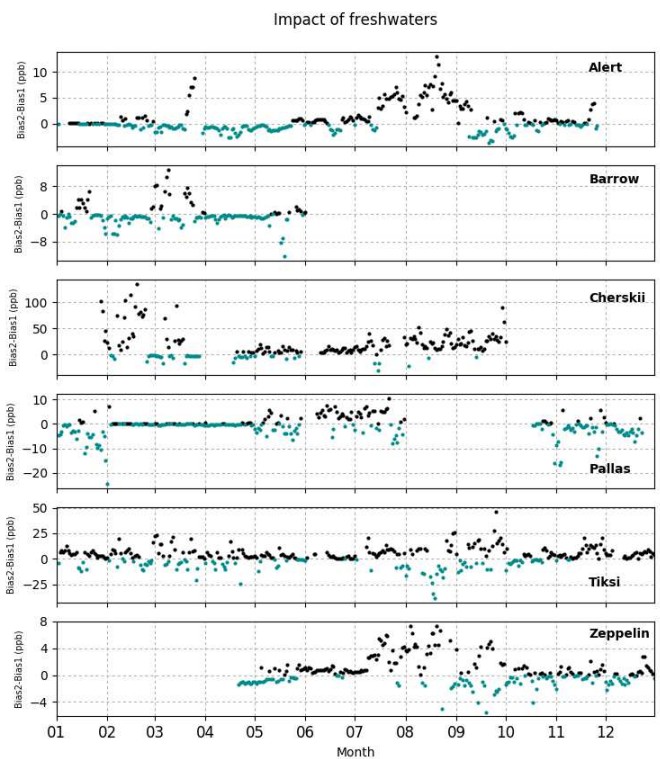

**Figure 9.** Difference between the absolute values of the biases between simulated and observed CH$_4$, for simulations using the two freshwater inventories, at six measurement sites, in 2012. Simulation 1 is the reference simulation. Simulation 2 includes the bLake4Me-derived lake emission inventory. Blue points indicate negative values. Note that different scales are used for each station.

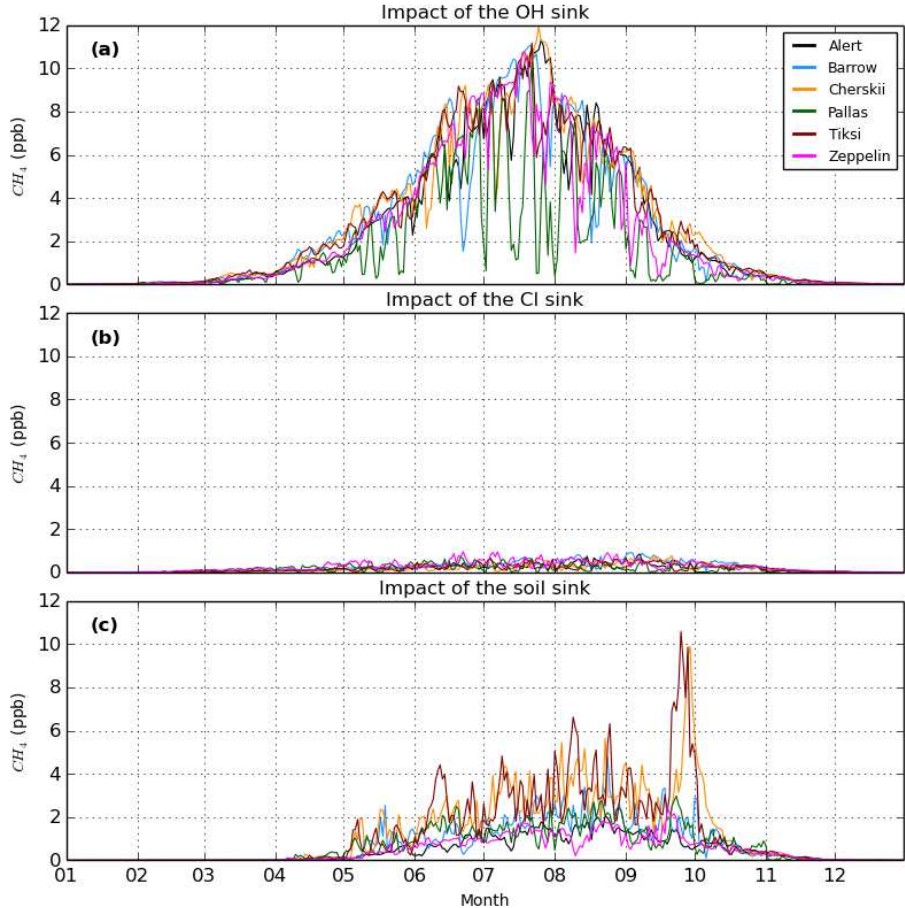

**Figure 10.** Difference between the reference simulation and (a) the simulation including the OH sink, (b) the one including the Cl sink, and (c) the one including soil uptake, at six measurement sites. Consequently, the impact of the sinks is shown here as positive values.

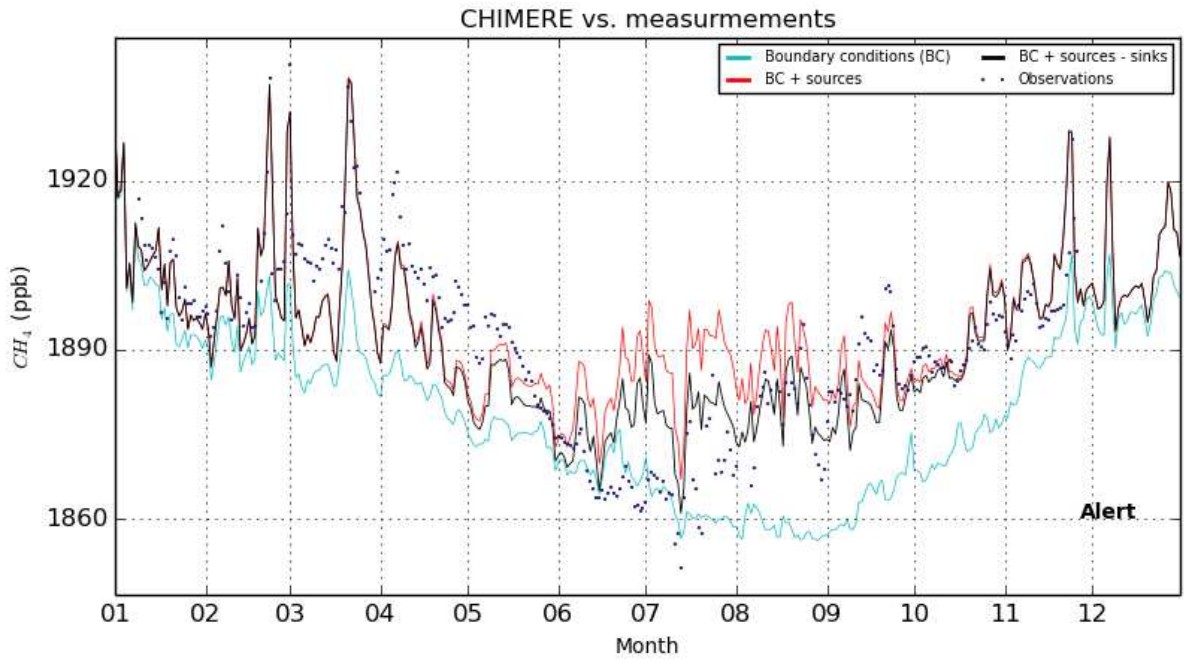

**Figure 11.** Time series of simulated and observed methane mixing ratios, at Alert, in 2012. The cyan line represents the contribution of the boundary conditions; the red line represents the added direct contribution of the

sources emitting in the domain; the black line includes the three added sinks (OH, soil, Cl). The blue points represent the observations. Time resolution for simulations and observations is 1 day.