# Peer review of "Detectability of Arctic methane sources at six sites performing continuous atmospheric measurements"

_Atmospheric Chemistry and Physics, 2017_

## Referee Comment (RC1) · Anonymous Referee #1 · 6 Apr 2017

General comments

This manuscript describes results of a modelling study of Arctic methane emissions using the CHIMERE chemical transport model. Simulated enhancements in methane from Arctic sources are compared with daily averaged methane mole fraction measurements from 6 sites for the year 2012. The impact of different sources at each of the 6 sites is quantified. Sensitivity tests have been carried out to compare different wetland and freshwater emissions and impact of methane sinks.

As the authors note Arctic methane emissions are uncertain and vulnerable to increase causing feedbacks as the region warms so the suggested improvements to fit emissions to observations are of importance. There have been other recently published regional model studies comparing methane model simulations with observations at Arctic stations (e.g. Warwick et al., 2016; Berchet et al., 2016 in Atmos. Chem. Phys.) but this manuscript has some new results such as consideration of freshwater emissions model and study of the effect of the methane sink, often not considered in regional scale methane models, which improves the correlation with observations.

The manuscript is written clearly with very few technical corrections required. The figures are also clear and discussion of the results is thorough.

I list below some comments and suggestions for minor modifications to the manuscript.

Specific comments

Line 79 and later. What is the status of the Poulter et al. (submitted) publication which is referred to several times? If this has not been published then some more detail will be required regarding the wetland emissions taken from that manuscript.

Introduction. It would be interesting to note the global and Arctic estimated methane emissions to give perspective to the size of emissions from this region.

Line 178. Why was the year 2012 chosen?

Line 189. Note (and perhaps give reasons for) also the long periods of missing data at Zeppelin, Pallas and Cherski.

Line 193. Why was just background data selected for Barrow and Pallas. Could you give details of the criterion used to filter the data? Were all data included for the other sites or were they filtered at all?

Line 236. Have you assumed anthropogenic emissions are constant all year? Is this realistic? Are emissions expected to be higher in the winter due to more emissions from fossil fuels for heating purposes? Would we expect seasonality in gas extraction in Russia?

Line 261. Does Orchidee include any emissions from wetlands in winter which according to Zona et al., 2016 may be significant?

Line 701. You could also bring in a discussion of Warwick et al., 2016 here. That paper found a closer agreement between modelled and measured methane mole fraction and isotopic composition at Arctic sites by delaying the seasonality in wetland emissions.

Table 1: Why don't Alert and Tiksi have both altitude and intake height? What do the numbers in that column refer to for those sites?

Technical corrections

Line 57. Schwietzke is misspelt.

Line 117. The 2.9 Tg CH4 yr-1 should be referred to as an estimated annual emission for the ESAS rather than a measured flux.

Line 152. Missing full stop at the end of this line.

Line 183. Earth System Research Laboratory (add the word Research)

Line 197. Integrated is misspelt.

Thompson et al. has now been published in Atmos. Chem. Phys. so this reference should be updated.

Figure 1: It would be helpful if some of the gridlines were labelled with longitudes and latitudes.

---

## Referee Comment (RC2) · Anonymous Referee #2 · 16 May 2017

This paper describes an Arctic version of the model CHIMERE, which has been used with tagged emissions of methane to diagnose the influence of different sources on 6 observational sites.

For me, the most interesting new result in this paper is the freshwater lakes inventory work, as this is a non-negligible source of methane that many models neglect. I think it would be good to make this clearer in the abstract. I think it would also be worth pulling out some figures to quantify how important the lakes are in the abstract, eg freshwater lakes account for 11-26% of the signal at your sites. I would also suggest that it would be useful for potential readers if this were reflected in the title of the paper as well, if you agree that this is the most important aspect of the paper.

Another interesting finding was that a later wetland seasonal cycle seemed to agree best with the observations. I think this is of interest as (a) we have many different wetlands emissions inventories and we want to know which is best to use in models, and (b) this agrees with recent observations from Zona and modelling from Warwick. So I think this would be good to highlight in the abstract.

Section 3.1.3 line 423, and line 678: When describing the seasonal cycle of methane in the Arctic, I would expect there to be lower methane in summer because of the presence of OH, compared to in the darkness of winter. In my mind, this outweighs the higher emissions of methane from wetlands in summer. I would see that as the main driver of the seasonal cycle over the whole Arctic, with any deviations from this attributed to some local influence eg from nearby wetland emissions. I am not sure I would attribute the seasonal cycle to transport from outside of the domain unless you had evidence to back this up. Even if you do have those numbers, isn't it the fact that the OH influence is acting in the midlatitudes too, so ultimately the transport into the boundary is related to the OH seasonal cycle anyway? I suggest that this section is revisited, with the OH seasonal cycle in mind.

Specific minor points:

Use methane or CH4 consistently throughout manuscript. Same with American/British spelling eg analyzes/analyses, vapor/vapour. Also, does Pole need a capital letter?

Line 58: There were two recent OH sink papers in PNAS, by Rigby et al and Turner et al. Maybe worth referencing here too. Dalsoren 2016 reference contains a typo.

The submitted Poulter reference is mentioned a few times. Unless this is published first, perhaps a good idea to mention the project name, so people might be able to look it up a bit easier.

Line 132: I think it should be "of emissions" not "on emissions"

Line 148: methane and Arctic should be the other way around

Line 192: please explain why you only use the background data here

Line 215: do you really mean forecasts, or do you mean analyses?

Line 218: Define LMDz

Line 241: define FAO

Line 281: Perhaps worth stating the resolution in km here as well.

Section 3.3: does bLake4Me stand for anything?

Line 560: I suggest adding "(black dots)" after "A positive value", as the colours confused me at first.

Line 587: the numbers here are confusing. I would say " The bias is improved from -6.4 to -6.0 ppb over the year"

Line 618/fig 10a: setting the sink to be a positive value is confusing. Consider changing this, or explaining it a little to make less confusing.

Line 701: Warwick at al 2016 also supports a delayed seasonal cycle in wetland emissions.

Fig 6 and 7: the quality when I printed these is not good. There are fuzzy areas, and it's hard to see the boundary conditions and the observations.

---

## Author Comment (AC1) · 6 Jun 2017

We thank Referee #1 for her/his fruitful comments and general appraisal of the manuscript. Here are our answers.

Specific comments

*Line 79 and later. What is the status of the Poulter et al. (submitted) publication which is referred to several times? If this has not been published then some more detail will be required regarding the wetland emissions taken from that manuscript.

—> Poulter et al. is still under review for minor revisions. The latest (minor) comments

have been addressed and the authors are waiting for the final decision of the editor. However, the model results have already been used in the Global Methane Budget synthesis (Saunois et al., 2016). Saunois et al. (2017, in review in ACPD) also use this ensemble and analyse some characteristics of the wetland emissions produced by these models. Note that the references for all process-based wetland models are already listed in Table 2. The "wetland part" of Section 2.3 has been reorganised to be more precise about the Poulter ensemble.

Line 274: "The version of ORCHIDEE used in this study comes from Poulter et al. (submitted) (see also Saunois et al. (2016)), like the ten other land surface models used for sensitivity studies (cf. section 3.2). Following Melton et al. (2013), net methane emissions have been computed under a common protocol; the models use the same wetland extent and climate forcings. Wetland area dynamics are based on global wetland datasets produced with the GLWD (Global Lakes and Wetlands Database), combined with SWAMPS (Surface WAter Microwave Product Series) inundated soils maps. The emissions from these ten other models range from 10.1 up to 58.3 TgCH4 yr-1"

A reference to Saunois et al. (2016), who describe the ensemble in more details, is also made in Section 3.2.

*Introduction. It would be interesting to note the global and Arctic estimated methane emissions to give perspective to the size of emissions from this region.

—> Thank you, this has been inserted in the second paragraph of the introduction.

Line 78: "The Arctic represents now about 4% of the global methane budget (23 vs. 568 TgCH4 yr-1 for 2012, according to Saunois et al. (2016)). This budget is lower than bottom-up estimates (range 37-89 TgCH4 yr-1, according to the review by Thornton et al. (2016b)), which are affected by large uncertainties. Although there is no sign of dramatic permafrost carbon emissions yet (Walter Anthony et al., 2016), thawing permafrost could double 21st century's Arctic methane budget and impact climate for centuries (Schuur et al. 2015)."

*Line 178. Why was the year 2012 chosen?

–> 2012 was chosen because it was the most recent year available to us in terms of computed wetland emissions, for the 11 wetland models used here. This explanation has been added in the last paragraph of the introduction.

Line 178: "The study focuses on 2012, since this is the most recent year for which wetland emissions are available for a set of models in a controlled framework."

*Line 189. Note (and perhaps give reasons for) also the long periods of missing data at Zeppelin, Pallas and Cherski.

–> This has been added in the manuscript.

Line 202: "Gaps in Cherskii (October-January), Pallas (August-mid-October), and Zeppelin (January-April) data are due to instrument issues."

*Line 193. Why was just background data selected for Barrow and Pallas. Could you give details of the criterion used to filter the data? Were all data included for the other sites or were they filtered at all?

–> To be consistent, we decided to remove the filters used for Barrow and Pallas and use all data for all sites.

Line 210: "All valid data from the sites are used in this study, with no filter applied."

One of the motivations of this paper was to look at the performances of the model at the sites. So, even though a data selection is crucial when using observations to invert the fluxes, in our case it is not necessary.

Table 6 and Fig. 6, 7 and 9 have been updated accordingly.

Please note that Tables 5 and 6 have been additionally slightly modified because of a mistake found in the calculation of the figures.

These changes do not alter our conclusions.

*Line 236. Have you assumed anthropogenic emissions are constant all year? Is this realistic? Are emissions expected to be higher in the winter due to more emissions from fossil fuels for heating purposes? Would we expect seasonality in gas extraction in Russia?

–> Yes, we assumed constant anthropogenic emissions. It is expected that emissions are in part correlated to household heating. However, we assume that anthropogenic emissions also happen in summer, following for example Berchet et al. (Biogesciences, 2015; see Fig. 6 and section 5.2.2). Maintenance and welling works taking place in Russia during summer cause methane seepages that can be of importance. In the absence of more precise information, we keep anthropogenic emissions constant all year round.

*Line 261. Does Orchidee include any emissions from wetlands in winter which according to Zona et al., 2016 may be significant?

–> ORCHIDEE does not include winter emissions, like the other wetland models. A sentence has been added in the conclusion concerning this issue in wetland emission models.

Line 729: "In subsequent modelling studies, if wetland emission models still have the same seasonality, ways to somehow force winter emissions should be considered."

*Line 701. You could also bring in a discussion of Warwick et al., 2016 here. That paper found a closer agreement between modelled and measured methane mole fraction and isotopic composition at Arctic sites by delaying the seasonality in wetland emissions.

–> Warwick et al. (2016) is indeed a good element for the discussion. It is now part of the conclusion:

Line 725: "The forward modelling study of Warwick et al. (2016) also reached the same conclusions. To better capture the seasonal cycle of methane, wetland emissions needed to start no sooner than June and peak between July and September. This

result was backed by isotopologues data that suggested large contributions from a biogenic source until October."

*Table 1: Why don't Alert and Tiksi have both altitude and intake height? What do the numbers in that column refer to for those sites?

–> The correct numbers have been added in Table 1.

Technical corrections

*Line 57. Schwietzke is misspelt.

–> This has been corrected.

*Line 117. The 2.9 Tg CH4 yr-1 should be referred to as an estimated annual emission for the ESAS rather than a measured flux.

–> Our sentence has been rephrased properly.

*Line 152. Missing full stop at the end of this line.

*Line 183. Earth System Research Laboratory (add the word Research)

*Line 197. Integrated is misspelt.

–> These mistakes have been corrected.

*Thompson et al. has now been published in Atmos. Chem. Phys. so this reference should be updated.

–> The reference has been updated.

*Figure 1: It would be helpful if some of the gridlines were labelled with longitudes and latitudes.

–> Figure 1 has been improved accordingly.

References :

Berchet et al.: Natural and anthropogenic methane fluxes in Eurasia: a mesoscale quantification by generalized atmospheric inversion, Biogeosciences, 12, 5393-5414, doi:10.5194/bg-12-5393-2015, 2015.

Saunois et al.: Variability and quasi-decadal changes in the methane budget over the period 2000-2012, Atmos. Chem. Phys. Discuss., doi:10.5194/acp-2017-296, 2017.

---

## Author Comment (AC2) · 6 Jun 2017

We thank Referee #2 for his/her comments and careful reading of the manuscript. Here are our answers.

*For me, the most interesting new result in this paper is the freshwater lakes inventory work, as this is a non-negligible source of methane that many models neglect. I think it would be good to make this clearer in the abstract. I think it would also be worth pulling out some figures to quantify how important the lakes are in the abstract, eg freshwater lakes account for 11-26% of the signal at your sites. I would also suggest that it would be useful for potential readers if this were reflected in the title of the paper as well, if

you agree that this is the most important aspect of the paper.

–> We thank the reviewer for this comment. Some more references to freshwaters have been included in the abstract to strengthen the new development about lake emissions:

Line 27: "A polar version of the CHIMERE chemistry-transport model is used to simulate the evolution of tropospheric methane in the Arctic during 2012, including all known regional anthropogenic and natural sources, in particular freshwater emissions which are often overlooked in methane modelling."

Line 38: "In particular, freshwaters play a decisive part in summer, representing on average between 11 and 26% of the simulated Arctic methane signal at the sites."

We think that the title assumes a balance as we try to address all Arctic emissions and prefer to keep it as it is.

*Another interesting finding was that a later wetland seasonal cycle seemed to agree best with the observations. I think this is of interest as (a) we have many different wetlands emissions inventories and we want to know which is best to use in models, and (b) this agrees with recent observations from Zona and modelling from Warwick. So I think this would be good to highlight in the abstract.

–> We agree. A sentence has been added in the abstract about this aspect of the wetland seasonal cycle.

Line 43: "The closest agreement with the observations is reached when using the two wetland models whose emissions peak in August-September, while all others reach their maximum in June-July. Such phasing provides an interesting constraint on wetland models which still have large uncertainties at present."

As said in our reply to Ref.1, a few words have been added in the conclusion about Warwick et al. (2016).

Line 725: "The forward modelling study of Warwick et al. (2016) also reached the

same conclusions. To better capture the seasonal cycle of methane, wetland emissions needed to start no sooner than June and peak between July and September. This result was backed by isotopologues data that suggested large contributions from a biogenic source until October."

*Section 3.1.3 line 423, and line 678: When describing the seasonal cycle of methane in the Arctic, I would expect there to be lower methane in summer because of the presence of OH, compared to in the darkness of winter. In my mind, this outweighs the higher emissions of methane from wetlands in summer. I would see that as the main driver of the seasonal cycle over the whole Arctic, with any deviations from this attributed to some local influence eg from nearby wetland emissions. I am not sure I would attribute the seasonal cycle to transport from outside of the domain unless you had evidence to back this up. Even if you do have those numbers, isn't it the fact that the OH influence is acting in the midlatitudes too, so ultimately the transport into the boundary is related to the OH seasonal cycle anyway? I suggest that this section is revisited, with the OH seasonal cycle in mind.

–> This is right, the seasonal cycle is mostly driven by OH. When it is written, in the manuscript, that air from outside the domain is the main driver of the seasonal cycle for some sites, it implicitly meant that it was ultimately due to the influence of OH, and not due to some seasonal pattern of transport. This point has been clarified in Section 3.1.3.

Line 444: "Although Arctic emissions are greater in summer, Alert, Pallas and Zeppelin have higher methane values in winter due to higher influence of air coming from lower latitudes, whose methane seasonal cycle is mostly driven by OH."

Specific minor points:

*Use methane or CH4 consistently throughout manuscript. Same with American/British spelling eg analyzes/analyses, vapor/vapour. Also, does Pole need a capital letter?

–> "CH4" has been replaced by "methane" and British spelling has been favoured. "Pole" refers here to the North Pole, so we think it requires a capital letter.

*Line 58: There were two recent OH sink papers in PNAS, by Rigby et al and Turner et al. Maybe worth referencing here too. Dalsoren 2016 reference contains a typo.

–> Thank you, the references have been added, and the typo corrected.

Line 61: "A number of different processes have been examined including changes in anthropogenic sources (Schaefer et al., 2016; Hausmann et al., 2016; Schwietzke et al., 2016), in natural wetlands (Bousquet et al., 2011; Nisbet et al., 2016, McNorton et al., 2016), or in methane lifetime (Dalsøren et al., 2016; Rigby et al., 2017; Turner et al., 2017)."

*The submitted Poulter reference is mentioned a few times. Unless this is published first, perhaps a good idea to mention the project name, so people might be able to look it up a bit easier.

–> Poulter et al. is still under review for minor revisions. The latest (minor) comments have been addressed and the authors are waiting for the final decision of the editor. However, the model results have already been used in the Global Methane Budget synthesis (Saunois et al., 2016). Saunois et al. (2017, in review in ACPD) also use this ensemble and analyse some characteristics of the wetland emissions produced by these models. Note that the references for all process-based wetland models are already listed in Table 2. The "wetland part" of Section 2.3 has been reorganised to be more precise about the Poulter ensemble.

Line 274: "The version of ORCHIDEE used in this study comes from Poulter et al. (submitted) (see also Saunois et al. (2016)), like the ten other land surface models used for sensitivity studies (cf. section 3.2). Following Melton et al. (2013), net methane emissions have been computed under a common protocol; the models use the same wetland extent and climate forcings. Wetland area dynamics are based on global wetland

datasets produced with the GLWD (Global Lakes and Wetlands Database), combined with SWAMPS (Surface WAter Microwave Product Series) inundated soils maps. The emissions from these ten other models range from 10.1 up to 58.3 TgCH4 yr-1."

A reference to Saunois et al. (2016), who describe the ensemble in more details, is also made in Section 3.2.

*Line 132: I think it should be "of emissions" not "on emissions"

*Line 148: methane and Arctic should be the other way around

–> This has been corrected.

*Line 192: please explain why you only use the background data here

–> To be consistent, we decided to remove the filters used for Barrow and Pallas and use all data for all sites.

Line 210: "All valid data from the sites are used in this study, with no filter applied."

*Line 215: do you really mean forecasts, or do you mean analyses?

–> In fact both forecasts and reanalyses are used here, thank you for pointing this out.

*Line 218: Define LMDz

–> We think it is useless to define the acronym (which just stands for the name of the institute (LMD) where the model was developed, and "with Zooming capability"), but we added a reference.

Line 230: "Initial and boundary concentrations come from optimized global simulations of the LMDZ general circulation model for 2012 (Locatelli et al., 2015)."

*Line 241: define FAO

–> This has been added. I also forgot to mention that BP statistics were used as well for the anthropogenic emission projections.

Line 253: "Given that the EDGARv4.2FT2010 emissions are not available for years after 2010, the 2010 values are used for 2012 for every sector but the ones for which FAO (Food and Agriculture Organization, http://www.fao.org/faostat/en/#data/) and BP (http://www.bp.com/) data are available (oil and gas production, fugitive from solid, enteric fermentation, and manure management)."

*Line 281: Perhaps worth stating the resolution in km here as well.

–> This has been added.

*Section 3.3: does bLake4Me stand for anything?

–> No it does not.

*Line 560: I suggest adding "(black dots)" after "A positive value", as the colours confused me at first.

–> Thank you, this has been added in the text.

*Line 587: the numbers here are confusing. I would say "The bias is improved from -6.4 to -6.0 ppb over the year"

–> Thank you, this has been fixed.

*Line 618/fig 10a: setting the sink to be a positive value is confusing. Consider changing this, or explaining it a little to make less confusing.

–> A sentence has been added in the legend of Fig. 10.

Line 1325: "Figure 10. Difference between the reference simulation and (a) the simulation including the OH sink, (b) the one including the Cl sink, and (c) the one including soil uptake, at six measurement sites. Consequently, the impact of the sinks is shown here as positive values."

*Line 701: Warwick at al 2016 also supports a delayed seasonal cycle in wetland emissions.

–> As said above, a short discussion on this paper has been inserted.

*Fig 6 and 7: the quality when I printed these is not good. There are fuzzy areas, and it's hard to see the boundary conditions and the observations.

–> These figures have been re-processed. We hope they are more easily understandable now.